# A Digital-Twin-Based Detection and Protection Framework for SDC-Induced Sinkhole and Grayhole Nodes in Satellite Networks

**Gongzhe Qiao** [1], **Yi Zhuang** [1,*], **Tong Ye** [1] and **Yuan Qiao** [2]

1   College of Computer Science and Technology, Nanjing University of Aeronautics and Astronautics, Nanjing 211100, China
2   Harbin Electric Power Bureau, STATE GRID Corporation of China, Harbin 150050, China
*   Correspondence: zy16@nuaa.edu.cn

**Abstract:** In the space environment, cosmic rays and high-energy particles may cause a single-event upset (SEU) during program execution, and further cause silent data corruption (SDC) errors in program outputs. After extensive research on SEU and SDC errors, it has been found that SDC errors in the routing program in satellite networks may lead to the emergence of Sinkhole (SH) and Grayhole (GH) nodes in the network, which may cause damage to satellite networks. To find and solve the problems in time, a digital-twin-based detection and protection framework for SDC-induced SH and GH nodes in satellite networks is proposed. First, the satellite network fault model under SEU and the generation mechanism of SH and GH nodes induced by SDC errors are described. Then, the data structure based on digital twins required by the proposed detection and protection framework is designed, and the detection methods of SH and GH nodes induced by SDC errors are proposed. SKT and LLFI simulation tools are used to build a simulated Iridium satellite network and carry out fault injection experiments. Experiment results show that the accuracy of the proposed detection method is 98–100%, and the additional time cost of routing convergence caused by the proposed framework is 3.1–28.2%. Compared with existing SH and GH detection methods, the proposed methods can timely and accurately detect faults during the routing update stage.

**Keywords:** satellite network; routing mechanism; SDC error; digital twin; sinkhole; grayhole; detection and protection

---



## 1. Introduction

According to data from the National Oceanic and Atmospheric Administration (NOAA) Space Environmental Service Center, single-event upset (SEU) events caused by high-energy particles occur every year on average [1]. SEU refers to the reaction caused by a single high-energy particle entering the sensitive area of semiconductor devices (such as microprocessors, semiconductor memories, or power transistors), resulting in a single-bit upset of memory cells [2]. In the complex space environment, cosmic rays or high-energy particles can cause SEU in satellite systems and cause errors in program outputs. Silent Data Corruption (SDC) is a type of SEU error that is hard to detect [3]. SDC errors in satellite routing programs may spread to the satellite network during the routing update process, causing the satellite network to be subject to various threats. Among them, Sinkhole (SH) and Grayhole (GH) nodes are among the most serious threats in the network layer.

In this study, we find that in the route discovery phase, a satellite affected by SDC errors may claim to have a false inter-satellite link, and the data packets passing through the link will be discarded, resulting in a SH node. In the routing planning stage, SEU may cause data loss, destination node change, and next hop node change in the routing table, which will make the affected satellite unable to find the next hop and discard the data

packets, thus generating a GH node. In this paper, we regard the nodes that discard all packets passing through themselves (Blackhole nodes) as a special case of GH nodes.

Digital twins (DTs) are virtual reflections of physical objects, which can make full use of the physical model, sensor update data, history data, and other information to map the real physical object in the virtual digital space [4]. The combination of digital twins and satellite networks is still in the exploration stage. The US Air Force has used a digital twin of the Lockheed Martin GPS IIR satellite to detect cybersecurity issues by performing penetration testing on the digital replica of the satellite. The satellite network topology is time-varying, and the satellite movement in orbit is predictable [5]. Therefore, most satellite network routing update mechanisms consist of two main stages: the route discovery phase and the route planning phase. Network threats have always been a research hotspot. The existing attack detection methods are mainly based on machine learning [6,7] and rules [8,9]. However, as shown in Figure 1, most of the above methods detect the traffic during the satellites' data forwarding process, after the routing update, and cannot effectively detect the fault node when the traffic does not pass through the problem node [10]. The proposed method works at the routing update stage. By dividing the routing update process into a route discovery phase and a route planning phase, a routing information base (RIB) generated during the route discovery phase could be completed before data forwarding, and the detection of routing tables may be completed before the satellite queries the routing table (data forwarding stage). For SHs and GHs induced by SDC errors, it is worthwhile to design a detection and protection framework that could complete the detection as soon as possible and occupy a small amount of satellite computing resources.

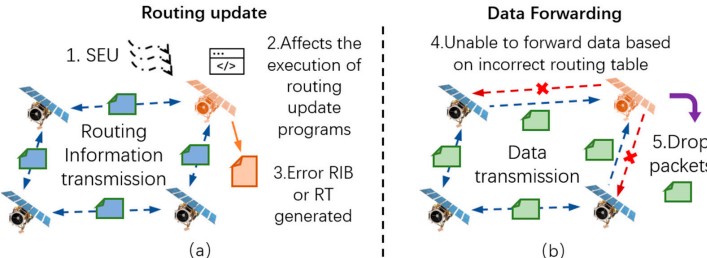

**Figure 1.** Comparison between the method proposed in this article and existing methods. (**a**) The proposed method works at this stage, which uses DT and process files to detect occurrences of SHs or GHs. (**b**) The existing SH or GH detection method usually works at this stage, which uses node behavior and network traffic.

The main challenges facing any detection and protection technology of SH and GH nodes induced by SDC errors are: (1) SHs and GHs are relatively serious threats on the network layer, and current research mainly focuses on SHs and GHs caused by malicious nodes. SEU can cause SDC errors in the satellite network routing programs, and further cause SHs and GHs. At present, few studies analyze the generation mechanism of the SH and GH nodes induced by SDC errors. (2) The existing research determines whether SH nodes and GH nodes exist in the satellite network by establishing a model based on rules or machine learning and analyzing the forwarding behavior or traffic of the satellites. For SH and GH nodes induced by SDC errors, such methods fail to detect whether SDC errors can cause SH and GH nodes promptly after SDC errors occur. (3) Digital twin technology can enable ground stations to obtain satellite network status information. At the same time, the ground station can judge whether the satellite network has problems through a detection algorithm. However, the above methods will also increase the additional computing and communication time costs of the satellite system. Therefore, a satellite system based on digital twins needs to consider the balance between detection efficiency and additional costs after the implementation of a detection and protection framework.

To solve the above problems, we propose a digital-twin-based detection and protection framework for SDC-induced SH and GH nodes in satellite networks. The contributions of this paper are as follows:

1. By carrying out fault injection experiments and SDC error analysis for each program, we establish a satellite network fault model under SEU. Also, we discuss the generation mechanism of SH and GH nodes induced by SDC errors by analyzing a typical satellite network routing mechanism and the behavior of satellites in each routing phase, providing theoretical support for subsequent detection and protection methods.

2. We propose a digital-twin-based detection and protection framework for SH and GH nodes induced by SDC errors. Before the actual data transmission, the proposed framework does all that it can to detect the SH and GH nodes induced by SDC errors in the satellite network and recovers the fault nodes, which provides technical support for the availability and reliability of the satellite network.

3. We propose a detection algorithm of SH and GH nodes based on digital twin routing data, which can complete the detection before data forwarding. In the simulated Iridium network environment, the experiment results show that the accuracy of the proposed detection method is 98–100%, and the additional time cost of routing convergence caused by the proposed framework is 3.1–28.2%.

The rest of this paper is organized as follows. The literature is reviewed in Section 2. The satellite network fault model under SEU and the generation mechanism of SH and GH nodes are presented in Section 3. The detection and protection framework for SDC-induced SH and GH nodes in satellite networks is proposed in Section 4. Section 5 shows the experimental evaluation results of the proposed method. And Section 6 concludes this paper.

## 2. Related Work

A number of research studies have applied digital twin technology to the design of satellite network protection frameworks and methods. For satellite security monitoring and verification, Hou et al. [4] present a framework that combines digital twins with runtime verification and propose a state synchronization method to ensure secure and efficient long-distance communication between satellites and digital twins. To monitor the satellite's behavior in real time and ensure the reliability of the satellite systems, Shangguan et al. [11] present a fault diagnosis and health monitoring (FD-HM) approach based on digital twins. For satellite–terrestrial networks, the satellite moving speed is faster than the ground station, which can cause inconsistent service and frequent satellite handover. To solve these problems, Zhao et al. [12] propose a digital-twin-assisted storage strategy for satellite–terrestrial networks (INTERLINK) and a satellite-storage-oriented handover scheme. The Iridium constellation is widely used as a satellite network simulation scenario. For example, Liu et al. [13] propose an intelligent energy-aware routing protocol for satellite networks and use the Iridium constellation as an experimental scenario for performance evaluation. At the same time, some scholars try to combine digital twins with industrial facilities [14–16].

The detection of SH and GH nodes has been a hot research topic with great concern. Many detection algorithms have recently been proposed in this area. To protect the EIoT environment against SH attacks, Pundir et al. [8] propose an intrusion detection scheme called SAD-EIoT, in which edge servers perform SH attack detection by exchanging messages. Prathapchandran et al. [17] propose a lightweight SH detection scheme RFTrust, which uses Random Forest (RF) and Subjective Logic (SL) to improve the efficiency of SH detection. Zaminkar et al. [18] propose a SH detection method SoS-RPL, which ranks the nodes in the network and allows child nodes to detect the malicious parent by using the routing graph information. Machine Learning (ML) can also be effective in detecting Blackhole attacks. Gao et al. [6] constructed a behavior classifier to detect Blackhole attacks in opportunistic networks. They also designed a collusion filtering strategy to improve detection accuracy. In another study [7], three supervised ML algorithms are trained and evaluated for detecting rank and Blackhole attacks in IoT networks. At the same time, there are also studies to improve network security by strengthening routing strategies [19–21]. Considering the periodic topology changes of satellite networks, Pan et al. [19] propose

OPSPF, which makes use of the regularity of constellation and performs periodic routing calculations for the generation of instantaneous routing tables.

In the process of routing update, the error output of the program may bring disastrous consequences to the satellite network. Therefore, there are some error detection and program protection methods [22–24]. To provide flexible and effective error protection, Didehban et al. [22] propose an instruction duplication error protection scheme gZDC, which can enhance protection capabilities by using coarse-grained scheduling and asymmetric control-flow signatures. So et al. [23] propose a compiler-level redundant multithreading scheme EXPERT, which can detect the manifestation of transient and permanent faults in hardware components. By adding redundant threads, they propose a software-level triply redundant multithreading scheme FISHER [24], which can improve error detection and recovery. However, the above error detection and program protection methods do not take into account the specific characteristics of satellite networks such as latency, limited bandwidth, and constrained resources. These constraints result in such methods not being well adapted to satellite networks. Especially under resource constraints, the redundancy mechanism of the above methods will greatly increase the computational burden and energy consumption of satellites. In the framework proposed in this article, detection and protection are carried out by ground stations, which greatly reduces the computational and storage pressure on satellites.

A comparison of the existing detection and protection methods is shown in Table 1. As far as we know, there are few methods for detecting SH and GH nodes caused by SDC during routing updates. Therefore, this article analyzes and compares traffic- and behavior-based SH or GH detection methods (TBDMs) and SDC error detection methods (SEDMs). TBDMs can determine the presence of SH or GH nodes based on the behavior of nodes or traffic information in the network. However, these methods work in the forwarding stage, and, therefore, they cannot detect SH or GH nodes caused by SDC (during the routing update stage) in time. In addition, TBDMs require satellites to run additional detection programs, which increases the computational burden and energy consumption of satellites. SEDMs can detect possible SDC errors through redundancy during program execution. However, these methods increase program execution time, satellite computational burden, and satellite energy consumption, and do not meet the high-efficiency requirements of routing programs. Also, SEDMs are prone to overprotection, namely the introduction of false alarms by detecting benign faults (faults that are going to be masked).

**Table 1.** Comparison of the existing detection and protection methods.

| | Methods | Description | Disadvantages |
|---|---|---|---|
| Traffic- and behavior-based detection and protection method (TBDM) | SAD-EIoT [8] | 1. Nodes perceive network status by exchanging messages. 2. Detect SH nodes in the network through assumptions and assertions. | 1. Working in the forwarding stage, and, therefore, cannot SH or GH nodes caused by SDC (during the routing update stage) in time. 2. Requiring one or more satellites with strong computing power. 3. Increasing computational burden and energy consumption of satellites. |
| | RFTrust [17] | 1. Nodes perceive network status by exchanging messages. 2. Random forest and subjective logic are used to detect SHs. | |
| | SoS-RPL [18] | 1. Nodes can exchange information with each other. 2. Child nodes can only detect the malicious parent by using the routing graph information. | |
| | CEBD [6]/ AutoML [7] | 1. Collect and analyze data exchanged between nodes. 2. Construct behavior classifiers to distinguish the blackhole behaviors from rational ones. | |

**Table 1.** *Cont.*

| | Methods | Description | Disadvantages |
|---|---|---|---|
| Error detection and protection method (SEDM) | EXPERT [23] | 1. Duplicate application main thread. 2. Main thread updates memory, while the other loads values from memory and detects errors. | 1. Increasing program execution time, which does not meet the high-efficiency requirements of routing programs. 2. Increasing computational burden and energy consumption of satellites. 3. Overprotection problem, namely introducing false alarms by detecting benign faults (faults that are going to be masked). |
| | FISHER [24] | 1. Triplicate application main thread. 2. Main thread updates memory and the redundant threads perform error detection. | |
| | gZDC [22] | 1. Duplicate arithmetic and logical operations. 2. Replicate the execution of critical instructions and check for errors by comparing the values of redundant register operands. | |
| DT-based detection and protection method | Our method | 1. Virtual satellite network updates according to the actual satellite network. 2. Check the routing update process file to determine if a soft error has occurred and could cause a SH or GH. | 1. Increasing satellite communication overhead. |

## 3. Fault Model and SH and GH Node Generation Mechanism

Transient faults or soft errors are usually caused by high-energy protons, electromagnetic interference, or galactic cosmic rays, and they are considered one of the most daunting reliability challenges for microprocessors [22]. Transient faults can lead to random bit flips in hardware devices such as registers, and may further cause application crashes, hangs, and incorrect running results [25]. Although a soft error will not cause hardware damage or loss, the application errors caused by it may also lead to catastrophic failure [26]. At present, the closest approach to the actual situation is to conduct irradiation experiments using a particle accelerator. However, this method is expensive and difficult to schedule. Software-based simulation fault injection methods (such as LLFI) are more flexible and cost-effective and the effectiveness of LLFI has been demonstrated in [27]. Therefore, we use LLFI for the fault injection campaign in this paper. Based on LLVM, LLFI is able to perform static analysis and inject faults in selected locations in the program. At the same time, LLFI can inject faults into the routing programs by modifying the source or destination register values of the targeted instruction, potentially leading to the generation of SDC. This capability aligns with the requirements of this article. LLFI allows users to select the instructions to be injected in the configuration file, and its default injection option is all instructions that can be injected. In this paper, we use the default injection option.

The main research object of this paper is the routing mechanism based on the Walker constellation architecture [28] and the time slot mechanism [29]. Through extensive research on SEU and SDC errors, we find that SEU may cause SDC errors in routing programs, and further cause SH and GH nodes in satellite networks, which will have a serious impact on the availability and reliability of satellite networks. Therefore, we conducted information collection and fault injection experiments on satellite routing programs at different phases. By analyzing the experimental data, we obtain the impact mode of SEU on routing programs, construct the fault model, and present the generation mechanism of SH and GH nodes.

### 3.1. Satellite Network Fault Model under SEU

Generally, most satellite network routing mechanisms include two main phases: the route discovery phase and the route planning phase. The main function of the route discovery phase is to collect a link-state advertisement (LSA), update the local network link-

state information, and generate a routing information base (RIB) for subsequent routing planning. In the routing planning phase, satellite nodes obtain all the link-state information of the satellite network from the RIB, use the path planning algorithm to plan the paths to other nodes in the satellite network, and generate routing tables.

SEU may cause four results in program execution: Masked, SW Detected, OS/HW Detected, and SDCs [22]. Masked means that the running result of the affected program is equal to the golden run result of the program. That is, the SEU has no impact on the output of the program. SW Detected represents cases where the protection scheme detects the manifestation of an error and raises the error detection flag. OS/HW Detected represents cases where fault injection simulation runs terminate permanently by generating an exception (i.e., segmentation faults or unknown instruction exception) or cause a time-out error. Based on whether it is a case of SW Detected or OS/HW Detected, the system or staff can make an appropriate response. The program in which an SDC error occurs can execute normally without any indication of system errors but may output error results, a situation that is difficult to detect [3]. Therefore, in this paper, we mainly focus on SDC errors (the changes in program execution results caused by fault injection) and the impact of their further propagation on the satellite network. We conduct 10,000 fault injection experiments for each program in the route discovery and planning phases. We note that some SDC errors that may be caused by fault injection belong to the same category. For example, in the fault injection scenario of the Build_RIB program, 173 SDC errors belong to the false links category (i.e., an unconnected path marked as connected). That is to say, even though SDC errors are different, these SDC errors may belong to the same category. In this context, we can achieve the desired results through 10,000 fault injection experiments. Through statistical analysis of the fault injection results, we find that SDC errors caused by SEU in satellite network routing programs are likely to cause the emergence of SH and GH nodes in the network.

Because the data stored in memory is usually protected by a checking mechanism (such as ECC) [30], we do not consider the impact of SEU on the static stored data. This paper mainly considers the program running errors caused by SEU in the computing unit of the processor. The proposed satellite network fault model under SEU is shown in Figure 2. The model is divided into two parts according to the impact of SEU on different routing phases.

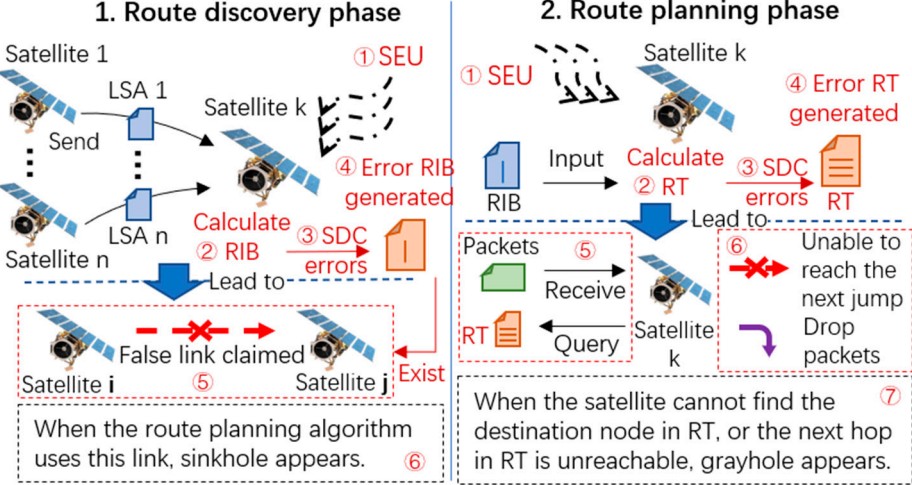

**Figure 2.** Satellite network fault model under SEU.

(1) In the route discovery phase, satellites can obtain the overall link states of the satellite network by sending and receiving link-state information from each other. Then satellites calculate and update the local RIBs. SEU may affect the satellite that computes the RIB, resulting in SDC errors in the generated RIB. One type of SDC error may cause false links in the RIB (Detailed in Section 3.2). In the subsequent route planning, the wrong RIB will be used as the input of the path planning algorithm, and further affect the satellite

node to make it believe that there is a false link. When the path planning algorithm adopts the false link, it can be considered that the satellite node is deceived and a SH node appears in the satellite network.

(2) In the route planning phase, the satellite calculates the optimal path through the path planning algorithm, obtains the next hop node ID, and generates a routing table. SEU may affect the execution of the path planning program, cause path planning errors, and further cause SDC errors in the generated routing table. SDC errors include data loss in the routing table, a change in destination nodes, or unreachable next hops (Detailed in Section 3.3). When any of the above SDC errors appear in the wrong routing table, it can be considered that there may be GH nodes in the satellite network. In the subsequent data forwarding process, the satellite node queries the wrong routing table, and may not be able to find the next hop and discard the data packet.

To explain the generation mechanism of the SH and GH nodes and the detection and protection framework proposed in this paper in detail, we construct a satellite network scenario based on the Iridium network structure [13] and DHRP [31]. DHRP is a typical routing protocol based on link delay. The programs executed by satellites in the route discovery phase and the route planning phase are shown in Table 2. Based on this scenario, we further discuss the impact of SEU on the satellite network routing mechanism and the generation mechanism of SH and GH nodes induced by SDC errors.

**Table 2.** The main programs executed by satellites during the routing phases.

|  | Route Discovery Phase | Route Planning Phase |
| --- | --- | --- |
| Name | Build_RIB | Dijkstra's Shortest Path (DSP) |
| Input | LSA (Link-state advertisement) | RIB (Routing information base) |
| Output | RIB | RT (Routing table) |
| Main function | Receive LSA information from other satellite nodes, obtain the overall link state of the satellite network, and generate an RIB. | Read the link-state information in the local RIB, use the Dijkstra algorithm to plan the shortest path to other nodes, and generate an RT. |

### 3.2. SH Node Generation Mechanism

The program call graph can describe the functional relationships in programs, which helps to understand program structure. Control-Flow Graphs (CFGs) are an abstract representation of all possible execution paths of the program flow, formed by linking basic blocks using directed edges (branches). Also, CFGs can depict the control flow execution of the corresponding function and the possible execution order of basic blocks and instructions. To explain the generation mechanism of the SH node, we take the program Build_RIB in the route discovery phase in Table 2 as an example to conduct fault injection experiments. First, based on the Build_RIB intermediate code generated by the Clang compilation, the program call graph is built in this paper, as shown in Figure 3. Build_RIB reads the LSA by calling the *readFile* function through the *main* function and writes the output results into the RIB by calling the *writeFile* function.

To simulate a SEU of the program, we use the LLFI tool [32] to inject 10,000 faults into the program in the form of a single-bit upset and carry out statistical analysis on the results. Potential SDC errors that may be caused by fault injection include format errors, link missing, weight changes, and false links. The error occurrence number and description are shown in Table 3. Among them, format error, link loss, and weight change cannot cause a SH node to form in subsequent routing planning. Also, if the false link delay is too long, the path planning algorithm will not use the link in the process of path planning, which thus cannot lead to a SH node forming. Therefore, we focus on the false link problem caused by SDC errors. By comparing the routing tables generated by the DSP with error RIBs and the golden run routing tables generated in the routing planning phase, we obtain 101 cases in which false links are used in the path planning. That is, SH nodes exist in the

satellite network and affect the network. In this way, we find that the probability of the occurrence of SH nodes that can affect the satellite network is about 1%.

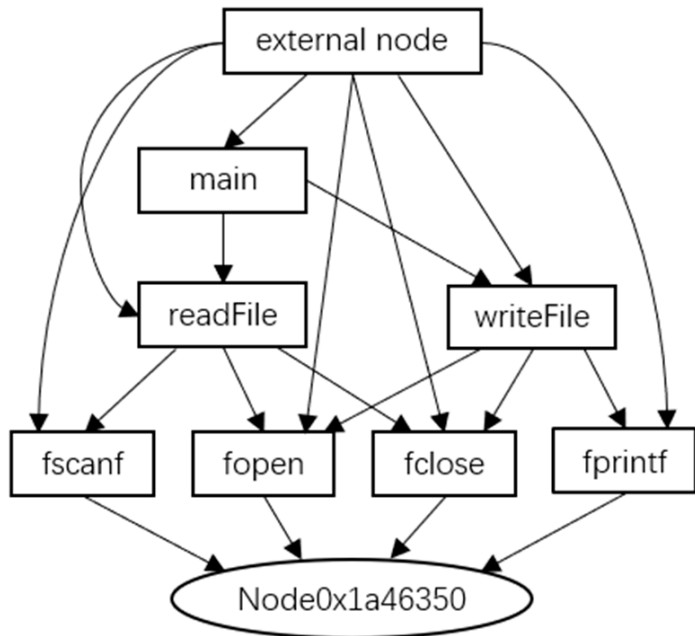

**Figure 3.** The call graph of Build_RIB.

**Table 3.** Build_RIB SDC error occurrence number and description.

| SDC Error | Occurrence | Description |
|---|---|---|
| Format error | 10 | SDC error causes the number of RIB columns to change. |
| Link missing | 124 | SDC error causes the original connected link to be marked as disconnected or missing. |
| Weight change | 121 | SDC error causes the link delay of the original connected path to change. |
| False link | 173 | SDC error causes an unconnected path to be marked as connected. We find that of these 173 false links, 101 can generate SH nodes. |

In the 101 cases where a SH node occurs, the instructions in the *readFile* function are affected 61 times, the instructions in the *writeFile* function 24 times, and the instructions in the main function 16 times. The reason why the instructions in the *readFile* function are affected 61 times is that during the Build_RIB program execution, the main function calls it 6 times. The *readFile* function has 20 basic blocks, among which the CFG and the instructions that lead to SH nodes forming are shown in Figure 4. The instructions that cause the most SH nodes are *load*, *icmp*, *sext*, *getelementptr*, and *add*. The *writeFile* function has 14 basic blocks, of which the basic blocks and the instructions that lead to SH nodes forming are shown in Figure 5. The instructions that cause SH nodes are mainly concentrated in the basic block %34, and the instructions that cause the most SH nodes are *load* and *getelementptr*. The main function has 55 basic blocks. Since there are six loop bodies with the same structure (Including seven basic blocks) in the *main* function, we map the instructions that lead to SH nodes forming in the last five loop bodies to the original loop body during the analysis. In this way, we obtain the CFG and the instructions that lead to SH nodes forming, as shown in Figure 6.

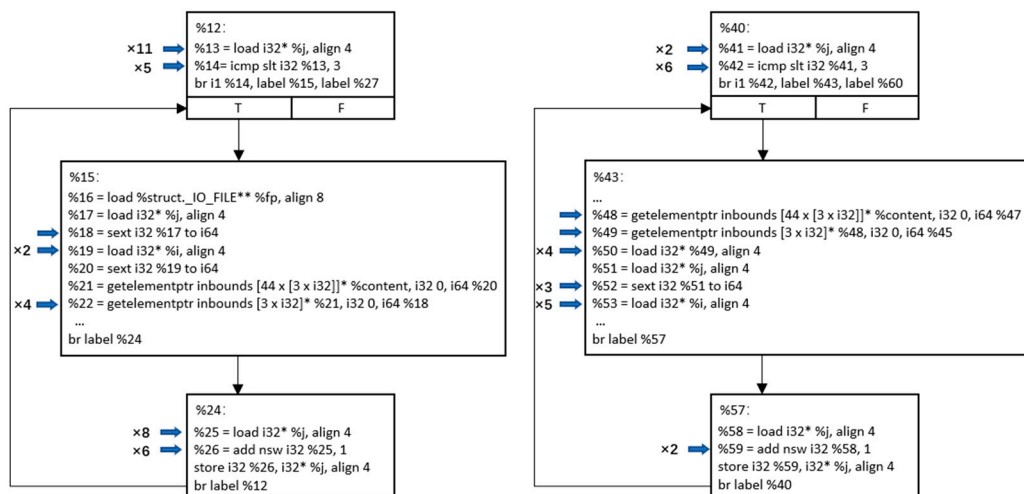

**Figure 4.** The CFG and the instructions in the *readFile* function that lead to SH nodes forming.

%34:
%35 = load %struct._IO_FILE** %fp, align 8
%36 = load i32* %j, align 4
%37 = sext i32 %36 to i64
...
%41 = getelementptr inbounds i32** %40, i64 %39
%42 = load i32** %41, align 8
%43 = getelementptr inbounds i32* %42, i64 %37
%44 = load i32* %43, align 4
...
br label %46

**Figure 5.** The basic blocks and the instructions in the *writeFile* function that lead to SH nodes forming.

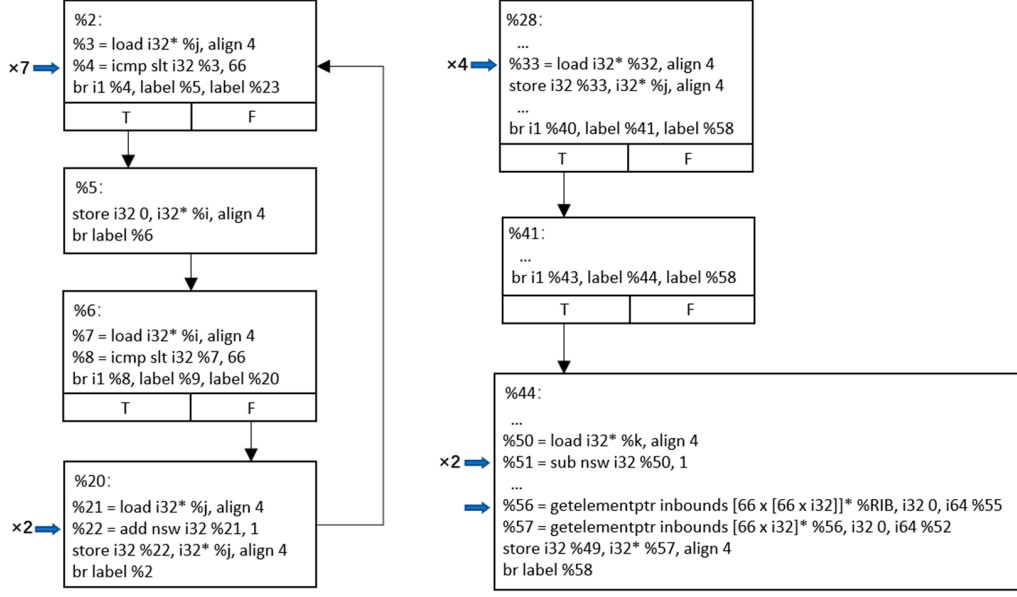

**Figure 6.** The CFG and the instructions in the *main* function that lead to SH nodes forming.

As shown in Figure 7, a typical example of a SH node caused by SEU is that during the dynamic cycle 61,316 of the Build_RIB program execution, the 12th bit of the *next* instruction (%18) of the basic block %15 in the *readFile* function is flipped, resulting in a false link between the No. 9 satellite node and the No. 56 satellite node. In the process of

path planning, because the link delay of this false link (Node 9 to 56) is 14 ms better than other paths, this false link is adopted by the path planning algorithm, and the next hop to reach Node 56 and its surrounding nodes in the generated routing table is Node 9. In the actual forwarding process, Node 9 cannot directly forward the data packet to Node 56. Therefore, Node 9 finds another link or drops the data packet, resulting in a SH in the satellite network.

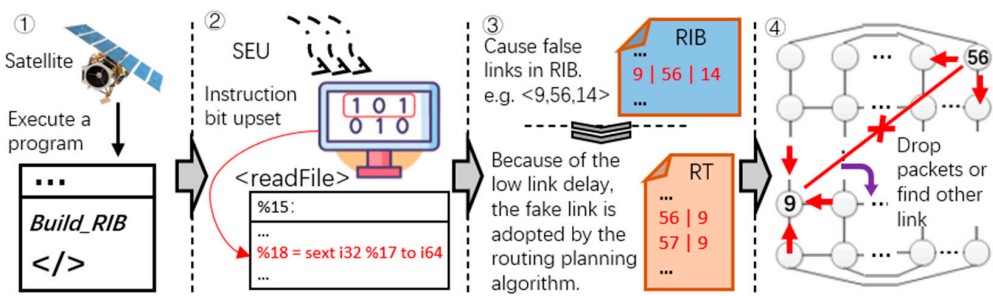

**Figure 7.** A typical example of a SH node caused by SEU.

### 3.3. GH Node Generation Mechanism

To explain the generation mechanism of GH nodes, we take the program Dijkstra's Shortest Path (DSP) in the route planning phase in Table 2 as an example to conduct fault injection experiments. First, based on the DSP intermediate code generated by the Clang compilation, the program call graph is built in this paper, as shown in Figure 8. First, DSP obtains the link-state information by calling the *fscanf* function through the *main* function. Then, DSP uses the *Dijkstra* function to plan the path from the satellite node to other nodes in the network and obtains the next hop node ID. Finally, the program generates a routing table in the format of the destination node ID and the next hop node ID.

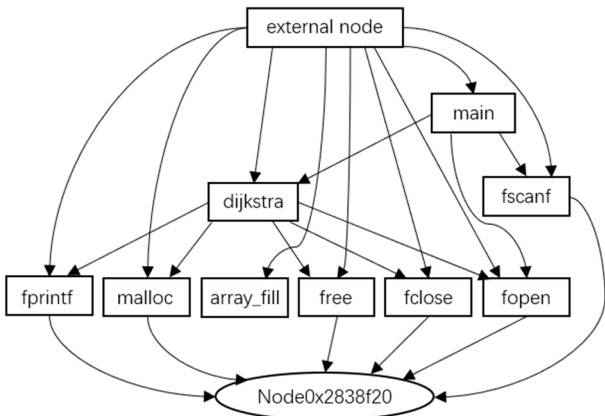

**Figure 8.** The call graph of Dijkstra's Shortest Path (DSP).

To simulate a SEU of the program, we use the LLFI tool to inject 10,000 faults into the DSP program in the form of a single-bit upset and carry out a statistical analysis of the results. We find that more than 80% of the results are Masked, SW Detected, or OS/HW Detected, and the number of SDC errors is 1173. Potential SDC errors that may be caused by fault injection include data loss, destination node changes, and next-hop node changes. The occurrence number and description of the errors are shown in Table 4. Because the influence of SEU on the DSP program ultimately affects the routing table, the data loss and the destination node change in the routing table can cause the satellite to fail to find the next hop, resulting in the emergence of a GH node in the satellite network. In most cases of next-hop changes, the next hop changes to other connectable neighbor nodes. Therefore, in the case of a next-hop change, we further obtain 188 cases where the next hop changes to a non-neighbor node, that is, the case where a GH node occurs in the satellite network.

In this way, we find that the probability of the occurrence of GH nodes induced by SEU is about 2%.

**Table 4.** DSP SDC error occurrence number and description.

| SDC Error | Occurrence | Description |
|---|---|---|
| Data loss | 9 | The route table misses one or more rows. The satellite fails to find the next hop of the corresponding destination node, which can generate a GH. |
| Destination node change | 5 | The destination node in the routing table changes. The satellite fails to find the next hop of the corresponding destination node, which can generate a GH. |
| Next-hop node change | 1159 | The next hop node in the routing table changes. A GH is generated when the next-hop node changes to a non-neighbor node, which occurs 188 times in total. |

GH nodes occur in 202 cases After statistical analysis, we find that in the nine cases of data loss, except for the *load* instruction (%10) of the basic block %8 in the *array_fill* function, the other seven cases occur in the %188, %192 and %253 basic blocks of the *Dijkstra* function. Meanwhile, the five cases of destination node change are caused by the *load* instruction (%201) in the basic block %197 of the *Dijkstra* function, as shown in Figure 9. At the same time, we find that in the 188 cases of next-hop node changes, in which GH nodes appear, the majority (161 times) occur in the *main* function, as shown in Figure 10, and the rest (27 times) occur in the *Dijkstra* function, as shown in Figure 11. Of these, the instructions in the *main* function that can cause GH nodes are mainly concentrated in the basic blocks %9, %12, %17, %33, and %37, and the instructions in the *Dijkstra* function that can cause GH nodes are mainly concentrated in the basic blocks of %28, %208, and %237.

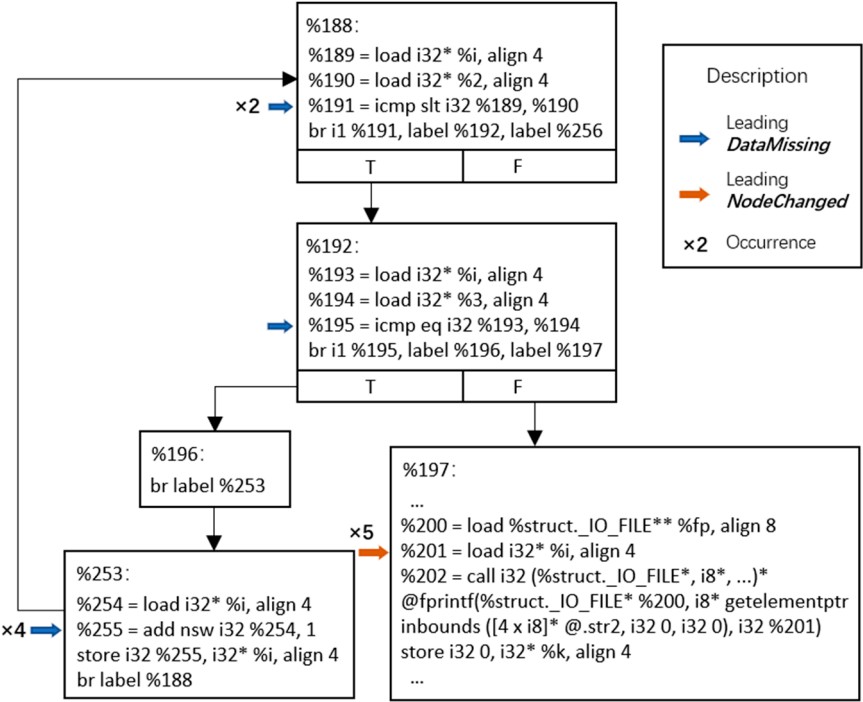

**Figure 9.** The CFG and the instructions in the *Dijkstra* function that lead to data loss and destination node changes.

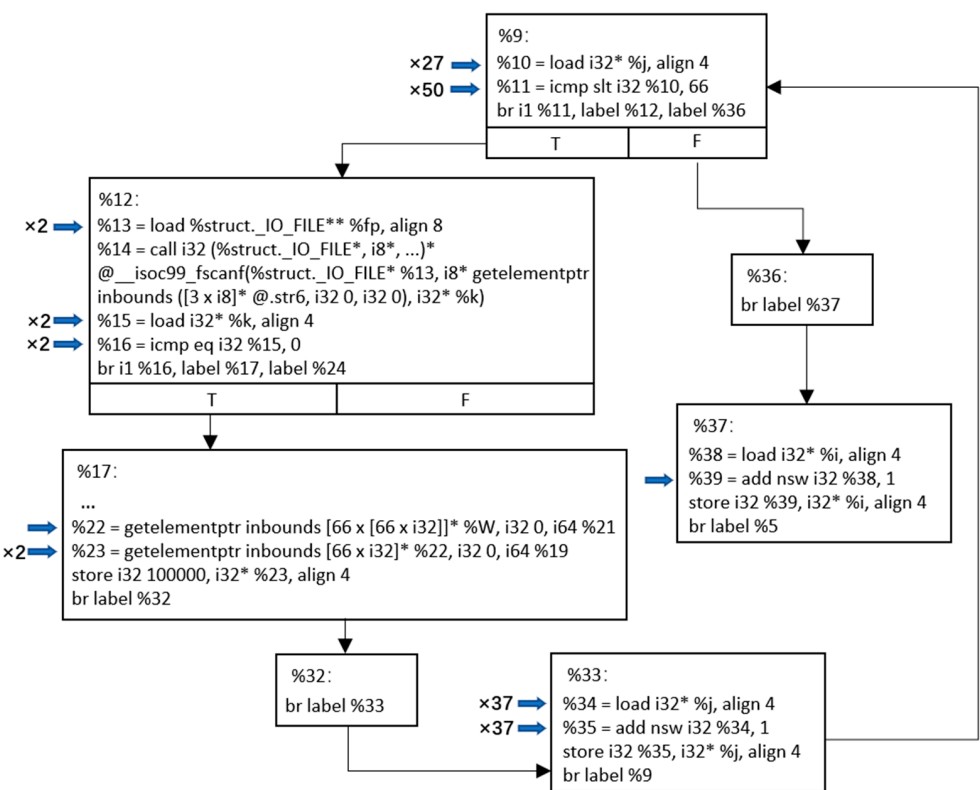

**Figure 10.** The CFG and the instructions in the *main* function that lead to next-hop node changes.

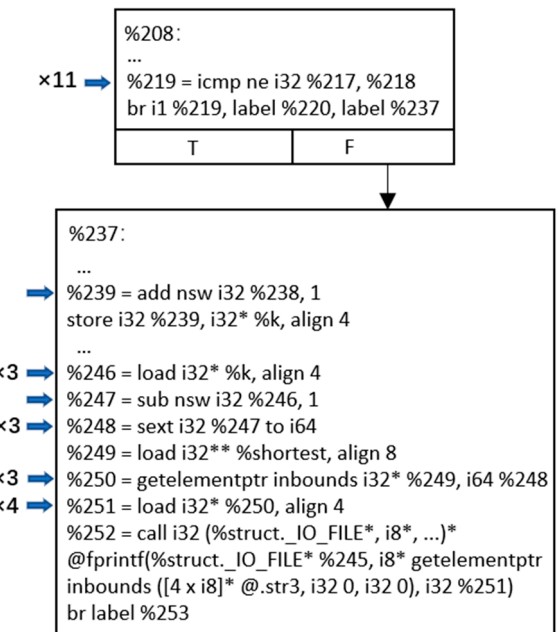

**Figure 11.** The CFG and the instructions in the *Dijkstra* function that lead to next-hop node changes.

An extreme case of a GH node is a Blackhole node, that is, a node that discards all packets passing through it. Blackhole nodes also exist in the above cases. As shown in Figure 12, a typical example of a GH node caused by SEU is that the No. 15 Satellite node is affected by SEU in the route planning phase. During dynamic cycle 12,618 of the DSP program execution, the 30th bit of the *add* instruction (%35) of the basic block %33 in the *main* function is flipped. As a result, all the next hops in the generated routing table are non-neighbor nodes and cannot be connected. Therefore, in the data forwarding phase,

node 15 could not forward the data packets to other nodes and finally dropped the data packets, resulting in a Blackhole in the satellite network.

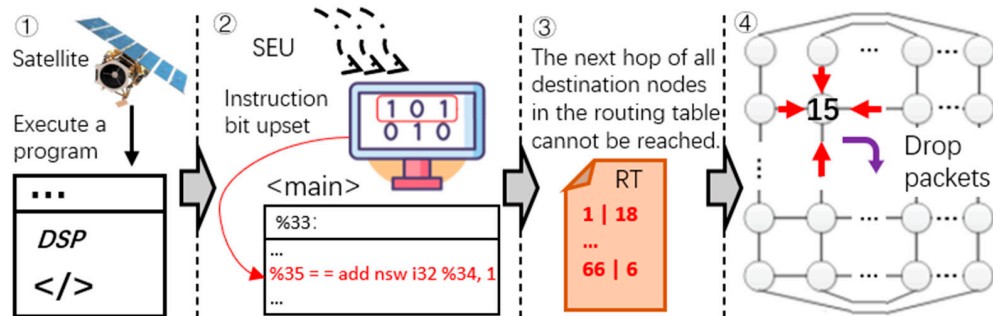

**Figure 12.** A typical example of a Blackhole node caused by SEU.

## 4. The Digital-Twin-Based Detection and Protection Framework

The combination of digital twins and runtime verification is still a very young idea [4]. Digital twins can make full use of the physical model, sensor update data, history data, and other information to map a real satellite network in a virtual digital space, thus reflecting the operation status of the satellite network. At the same time, the ground station staff can also find problems in the satellite network and make adjustments in time. In this section, we first define the digital-twin information in the proposed framework. Then we propose the detection and protection framework and describe the overall process and detection algorithm in detail.

### 4.1. Data Description in the Framework

At present, the existing state-of-the-art research in the field of satellite networks mostly employs two constellation designs for satellite networks: Walker delta (Inclined constellation) and Walker star (Polar orbit constellation). The satellite network structure based on the Walker constellation can be expressed as $N \times M/N/F(F = 0, 1, \ldots, N-1)$ [28], where $M$ is the number of satellites in a single orbit, $N$ is the number of orbits, and $F$ is the phase factor. Due to the predictability of satellite orbit and the stability of inter-satellite links, the routing mechanism combined with time slot is widely used in satellite networks [29]. During one time slot, the satellite network topology can be regarded as static. Satellites can obtain the current satellite network operation status and link states through routing discovery. As shown in Figure 13, we set the orbit ID $ID_{orb}$ in the satellite network topology from left to right to 1 to $N$, and set the satellite ID in the orbit $ID_{sat}$ from top to bottom to 1 to $M$. Then the satellite node ID in the satellite network can be expressed as $ID_{orb} \parallel ID_{sat}$. For example, the ID of the 6th node in the first orbit under the Iridium network structure is 106.

The detection and protection framework proposed in this paper needs some necessary information to synchronize and manage the satellite network. The information collected in actual satellite networks can be expressed as $I_p$, as shown in Equation (1), where $Pos_p$ represents the position information of satellites, $LSA_p$ represents the link-state advertisements of satellites, $RIB_p$ represents routing information bases, and $RT_p$ represents routing tables. In some plane-speaker-based or centralized planning algorithms, only the plane speakers or centralized controllers build the routing information base. Therefore, it is necessary to obtain K routing information bases based on the specific algorithms.

$$I_p = \begin{pmatrix} Pos_p \\ LSA_p \\ RIB_p \\ RT_p \end{pmatrix} = \begin{pmatrix} Pos_p^1, & Pos_p^2, & \ldots, & Pos_p^{N \times M} \\ LSA_p^1, & LSA_p^2, & \ldots, & LSA_p^{N \times M} \\ RIB_p^1, & RIB_p^2, & \ldots, & RIB_p^K \\ RT_p^1, & RT_p^2, & \ldots, & RT_p^{N \times M} \end{pmatrix} \tag{1}$$

The position information of satellite $i$ can be expressed as $Pos_p^i$, as shown in Equation (2), where $ID_{orb}^i$ represents orbit ID, $ID_{sat}^i$ represents satellite ID in the orbit, $LON^i$ repre-

sents the longitude of the satellite $i$, $LAT^i$ represents the latitude of the satellite $i$, and $ALT^i$ represents the altitude of the satellite $i$.

$$Pos_p^i = < ID_{orb}^i, ID_{sat}^i, LON^i, LAT^i, ALT^i > \qquad (2)$$

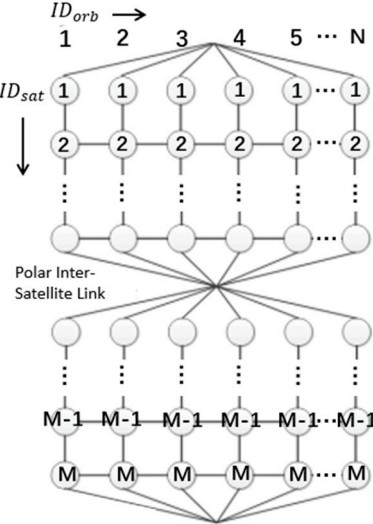

**Figure 13.** Topology diagram of satellite networks.

The link-state advertisements of satellite $i$ can be expressed as $LSA_p^i$, which is a set of link delay information between itself and neighbor nodes, as shown in Equation (3), where $ID_{orb}^j \parallel ID_{sat}^j$ represents the satellite node ID of satellite $j$, $DLY^{ij}$ represents the link delay between satellite $i$ and satellite $j$, and $i.NBR$ represents the neighbor nodes of satellite $i$. At the same time, the routing information base can be expressed as $RIB_p$, which is a set of $LSA$ of all nodes in the satellite network, as shown in Equation (4).

$$LSA_p^i = \left\{ < ID_{orb}^i \parallel ID_{sat}^i, \; ID_{orb}^j \parallel ID_{sat}^j, DLY^{ij} > \; | \; j \in i.NBR \right\} \qquad (3)$$

$$RIB_p = \left\{ LSA_p^k \; | \; k \in (0, N \times M) \right\} \qquad (4)$$

The routing table of satellite $i$ can be expressed as $RT_p^i$, as shown in Equation (5), where $DEST^k$ represents the destination node and $NEXT^k$ represents the next-hop node. Both $DEST^k$ and $NEXT^k$ can be expressed in the form of $ID_{orb} \parallel ID_{sat}$.

$$RT_p^i = \left\{ \left\langle DEST^k, NEXT^k \right\rangle | \; k \in (0, N \times M) \; \wedge k \neq i \right\} \qquad (5)$$

Similarly, the digital-twin information $I_{DT}$ obtained by the ground station also has the same data structure, as shown in Equation (6).

$$I_{DT} = \begin{pmatrix} Pos_{DT} \\ LSA_{DT} \\ RIB_{DT} \\ RT_{DT} \end{pmatrix} = \begin{pmatrix} Pos_{DT}^1, & Pos_{DT}^2, & \dots, & Pos_{DT}^{N \times M} \\ LSA_{DT}^1, & LSA_{DT}^2, & \dots, & LSA_{DT}^{N \times M} \\ RIB_{DT}^1, & RIB_{DT}^2, & \dots, & RIB_{DT}^K \\ RT_{DT}^1, & RT_{DT}^2, & \dots, & RT_{DT}^{N \times M} \end{pmatrix} \qquad (6)$$

*4.2. Detection and Protection Framework*

From the fault model described in Section 3, it can be seen that SEU can affect the execution of routing programs in satellite networks, and lead to the emergence of SH and GH nodes in satellite networks, thereby reducing the reliability and availability of satellite networks. Digital twins can reflect the operation status of the satellite network

through sensor data, satellite-ground communication data, and other information, which is convenient for detecting the abnormal status of a satellite network. Therefore, we propose a detection and protection framework for SH and GH nodes induced by SDC errors based on digital twins, as shown in Figure 14. The framework is divided into two main parts: the physical twin and the digital twin. The physical twin collects and monitors necessary data, such as satellite position and LSA. At the same time, the digital twin can enable the ground staff to obtain the operation status of the satellite network, find out the problems of the satellite network, and make adjustments in time. In addition, the secure communication protocol [4] is used to maintain communication and state updates between the digital twin and the physical twin. By mapping the process files generated by the satellite routing mechanism to digital space, the framework can do all that it can to find the SH and GH nodes before the satellites transmit data.

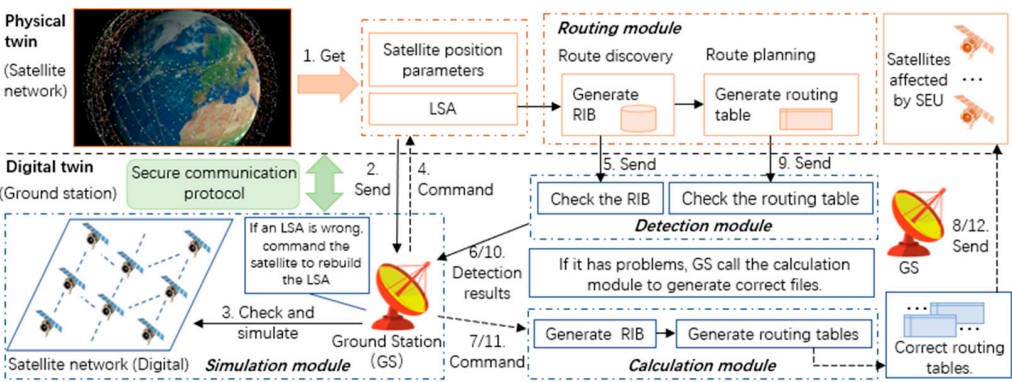

**Figure 14.** The detection and protection framework based on digital twins.

The proposed framework has four main modules: simulation module, routing module, detection module, and calculation module. The main functions of the simulation module include receiving LSA and satellite position information and updating the digital satellite network. The main functions of the routing module include route discovery and route planning. The RIB and RT generated in this model are important files for satellite routing. The main functions of the detection module include RIB detection and routing table detection. If the framework finds problems that can cause SH nodes or GH nodes, it will call the calculation module. The main functions of the calculation module include generating RIBs and RTs, which are used to recover the satellite network after finding problems.

After collecting LSA and satellite position information, the satellite network will send them to the ground station. The ground station uses the simulation module to update the digital satellite network. At the same time, the satellite network continues to execute the routing module, generate RIB and RT in stages, and send them to the ground station, respectively. It should be noted that the RIBs generated during the route discovery phase are directly sent back to the ground station (GS) without waiting for the route planning phase to complete. Therefore, after receiving the RIBs (5. in Figure 14), the GS directly calls the detection module to detect the RIBs and returns the detection result (6. in Figure 14). If the GS finds an RIB with a problem, it will call the calculation module (7. in Figure 14), calculate the RT for the affected satellite, and send it to the affected satellite (8. in Figure 14). Similarly, after receiving the RTs (9. in Figure 14), the GS calls the detection module to detect the RTs and return the detection result (10. in Figure 14). If the GS finds an RT with a problem, it will call the calculation module (11. in Figure 14), calculate the RT for the affected satellites, and send it to the affected satellites (12. in Figure 14). In the simulation scenario of this article, the faulty nodes (including SH, GH, and affected nodes) cannot find the correct next hop for forwarding data due to the wrong routing tables. Therefore, the framework proposed in this paper recovers the faulty nodes by sending the correct routing tables to the faulty nodes.

The framework proposed in this paper is mainly used to detect and protect a satellite network during its routing update process. The framework does not interfere with the normal routing behavior and has no additional computational overhead except for sending data. In addition, during RIB and routing table transmission, the satellite network does not need to wait for the ground station detection results and continues to perform the next phase or other tasks. According to data from NOAA [1], the probability of high-energy protons or galactic cosmic rays causing an SEU is 16%. Under the experimental settings of this article, the experimental results of this article show that the probabilities of SH or GH occurrence caused by SEU are around 1% and 2%, respectively. All measurements and calculations in Sections 3.2 and 3.3 are specific to the experimental settings of this article. In most cases, the satellite network can complete routing updates without the help of the ground station. When SEU causes SH nodes or GH nodes, the ground station can also detect and repair the routing tables in time. Therefore, compared with the traffic-based detection method, the framework proposed in this paper can complete the detection before data forwarding, thus greatly improving the reliability and availability of satellite networks.

### 4.3. The Detection and Protection Method in the Framework

In this section, we describe the overall workflow of the proposed detection and protection framework and describe the detection algorithms of SH and GH nodes in detail.

#### 4.3.1. Overall Workflow

Two assumptions are made in this paper: (1) the sensor equipment of the satellite is reliable; and (2) the credibility of the interaction information between the satellite and ground can be guaranteed through the secure communication protocol. The overall workflow of the detection and protection framework proposed in this paper is shown in Figure 15 and is divided into the following main steps.

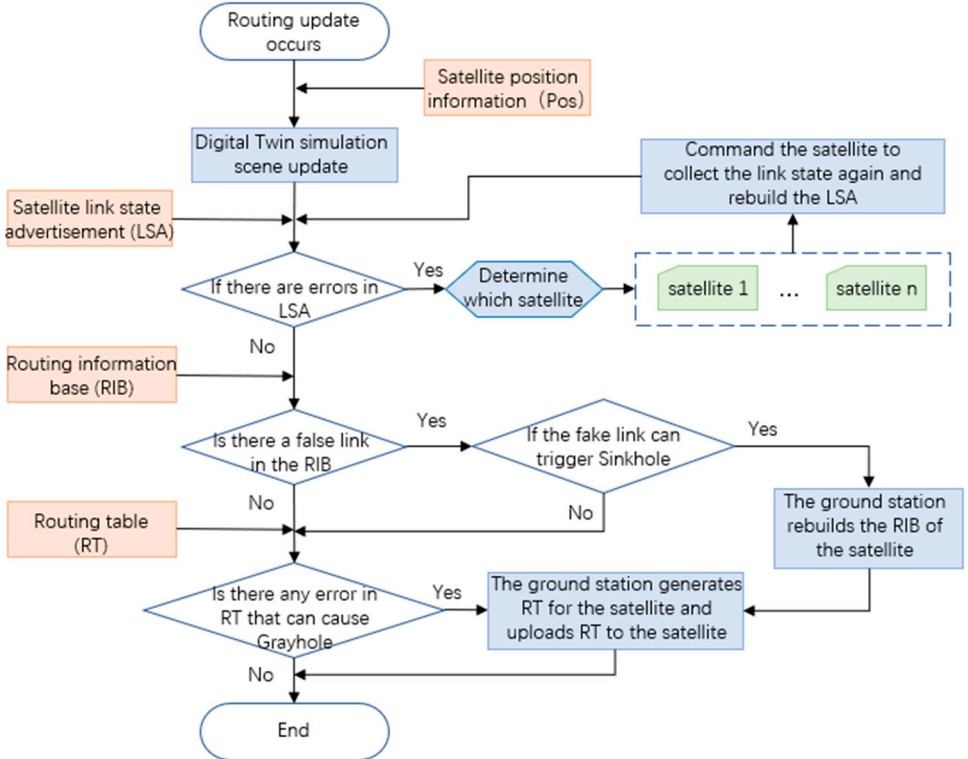

**Figure 15.** The overall workflow of the detection and protection framework.

Step 1. When the satellite network starts the routing update process, the satellite nodes obtain the current position information (Pos) and link-state advertisements (LSAs) through

the sensor and send them to the ground station. The ground station updates the digital simulation scenario according to the information received.

Step 2. After receiving the LSAs, the ground station detects whether the LSA has errors. If an LSA has errors, the ground station will further determine which satellite has errors, send commands to the satellite to rebuild the LSA, and let the satellite return the new LSA to the ground station.

Step 3. After routing discovery, the satellite network sends the RIBs to the ground station. Based on the digital-twin routing data, the ground station detects whether the RIB has a false link and further determines whether the false link can generate a SH node. If a SH node can be generated, the ground station will rebuild the RIB for the satellite, generate a routing table, and upload the routing table to the satellite.

Step 4. When sending RIBs, the satellite network does not need to wait for the detection results of the ground station and directly calculates the routing tables. Then, the satellite network will send the generated routing tables to the ground station for detection. If there is a routing table that can trigger a GH node, the ground station will regenerate the routing table for the satellite and upload it to the satellite.

### 4.3.2. SH Detection and Protection Method

Based on the SH generation mechanism proposed in Section 3.2, we design and implement a SH detection algorithm for the satellite network based on the Walker constellation and a routing mechanism based on link delay, as shown in Algorithm 1. After the satellite network generates an RIB, the ground station analyzes the adjacency of satellite nodes in the satellite network topology and the digital twin data $Pos_{DT}$, $LSA_{DT}$, and $RIB_{DT}$ to detect whether there is a false link that can trigger a SH. The proposed algorithm is as follows:

Step 1. (Line 1–2) Based on the data structure shown in Equations (3) and (4), for each piece of data $< ID_{orb}^{i} \parallel ID_{sat}^{i}, ID_{orb}^{j} \parallel ID_{sat}^{j}, DLY^{ij} >$ in $RIB_{DT}$, the algorithm firstly judges whether both satellites of the communication link are in the same orbit. If they are in the same orbit, jump to Step 2; If they are not in the same orbit, jump to Step 4.

Step 2. (Line 3–4) If the two satellites are in the same orbit, the algorithm further determines whether the two satellites are adjacent. If they are adjacent, this means there is an actual direct link. If they are not adjacent, this indicates that there is a false link in the satellite network, and it is necessary to further judge whether the false link is adopted, and jump to Step 3.

Step 3. (Line 5–9) Calculate the minimum number of hops $L_n$ in the orbit between two satellites, and judge whether $DLY^{ij}$ less than $L_n \times$ average delay $\overline{DLY_{iol}}$ of inter-orbital links (IOLs). If the condition is satisfied, this means that the false link is adopted and there is a SH node. If the condition is not satisfied, this means that the false link has not been adopted, and the SH node has not been generated. Continue to the next data.

Step 4. (Line 10) If the two satellites are not in the same orbit, the algorithm further determines whether the two satellites are in adjacent orbits and adjacent to each other. If they are adjacent, jump to Step 5; if they are not adjacent, jump to Step 7.

Step 5. (Line 11,15) Judge whether $LAT^{i}$ or $LAT^{j}$ is bigger than the polar region boundary latitude. If the condition is satisfied, this means that there is a false link, and jump to Step 6 to further judge whether the false link is adopted. If the condition is not satisfied, this means that there is an actual direct link, and this piece of data will not cause a SH node. Continue to the next piece of data.

Step 6. (Line 12–14) Judge whether $DLY^{ij}$ is less than $2 \times$ average delay $\overline{DLY_{iol}}$ of IOLs + average delay $\overline{DLY_{aol}}$ of intra-orbital links (AOLs). If the condition is satisfied, this means that the false link is adopted and a SH node exists. If the condition is not satisfied, this means that the false link has not been adopted, and the SH node has not been generated. Continue to the next piece of data.

Step 7. (Line 16–19) If the two satellites are not adjacent, this means that the link is a false link. The algorithm further judges whether the false link can be adopted. First, the

algorithm calculates the minimum number of hops $L_n$ in the orbit and the minimum number of hops $L_o$ between orbits. Then the algorithm judges whether $LAT^i$ or $LAT^j$ is bigger than the polar region boundary latitude. If the condition is satisfied, $L_n = \left| ID^i_{sat} - ID^j_{sat} \right| + 2$ and $L_o = \left| ID^i_{orb} - ID^j_{orb} \right|$; If the condition is not satisfied, $L_n = \left| ID^i_{sat} - ID^j_{sat} \right|$ and $L_o = \left| ID^i_{orb} - ID^j_{orb} \right|$. Jump to Step 8.

Step 8. (Line 20–22) The algorithm also judges whether $DLY^{ij}$ is less than $L_n \times \overline{DLY_{iol}} + L_o \times \overline{DLY_{aol}}$. If the condition is satisfied, this means that the false link is adopted and a SH node exists. If the condition is not satisfied, this means that the false link has not been adopted, and the SH node has not been generated.

---

**Algorithm 1** SH detection algorithm

---

Input: $Pos_{DT}$, $LSA_{DT}$ and $RIB_{DT}$
Output: if_Sinkhole_exist
Start

1.     For each piece of data $< ID^i_{orb} \parallel ID^i_{sat}, \; ID^j_{orb} \parallel ID^j_{sat}, DLY^{ij} >$ in $RIB_{DT}$
2.     if $ID^i_{orb} = ID^j_{orb}$:
3.         if abs$(ID^i_{sat} - ID^j_{sat}) = 1$ or abs$(ID^i_{sat} - ID^j_{sat}) = M - 1$:
4.            Not SH item, continue
5.         else:
6.            $L_n = \mathrm{Min}\left( \left| ID^i_{sat} - ID^j_{sat} \right|, M - \left| ID^i_{sat} - ID^j_{sat} \right| \right)$
7.            if $DLY^{ij} \leq L_n \times \overline{DLY_{iol}}$ :
8.                SH exists, break
9.            else: Not SH item, continue
10.    elif abs$(ID^i_{orb} - ID^j_{orb}) = 1$ and $ID^i_{sat} = ID^j_{sat}$:
11.         if $LAT^i$ and $LAT^j \geq$ Polar region boundary latitude:
12.            if $DLY^{ij} \leq 2 \times \overline{DLY_{iol}} + \overline{DLY_{aol}}$:
13.                SH exists, break
14.            else: Not SH item, continue
15.         else: Not SH item, continue
16.    else:
17.         if $LAT^i$ and $LAT^j \geq$ Polar region boundary latitude:
18.            $L_n = \left| ID^i_{sat} - ID^j_{sat} \right| + 2, L_o = \left| ID^i_{orb} - ID^j_{orb} \right|$
19.         else: $L_n = \left| ID^i_{sat} - ID^j_{sat} \right|, L_o = \left| ID^i_{orb} - ID^j_{orb} \right|$
20.         if $DLY^{ij} \leq L_n \times \overline{DLY_{iol}} + L_o \times \overline{DLY_{aol}}$:
21.            SH exists, break
22.         else: Not SH item, continue
23.    Detect the next piece of data
24.    return if_Sinkhole_exist

End

---

After the SH detection process, if an RIB that can cause a SH is detected, the ground station will use the satellite position information $Pos_{DT}$ and link-state information $LSA_{DT}$ to generate the correct RIB. Then the ground station generates a correct routing table for the satellite corresponding to the wrong RIB and uploads the routing table to the satellite. If there are no RIBs with errors, the ground station and satellite network require no additional operation.

### 4.3.3. GH Detection and Protection Method

Based on the GH generation mechanism proposed in Section 3.3, we design and implement a GH detection algorithm for the satellite network based on the Walker constellation and the routing mechanism based on link delay, as shown in Algorithm 2. After the satellite

network generates routing tables, the ground station analyzes the adjacency of satellite nodes in the satellite network topology and the digital twin data $Pos_{DT}$ and $RT_{DT}$ to detect whether there is an SDC error that can trigger a GH. The proposed algorithm proposed is as follows:

Step 1. (Line 1,13) Based on the data structure shown in Equation (5), for the digital twin routing table $RT^i_{DT}$ of satellite $i$, the algorithm firstly judges whether it contains all the satellite nodes except itself. If the condition is satisfied, the algorithm jumps to Step 2 to further judge whether the next hop is reachable. If the condition is not satisfied, this means that satellite $i$ cannot find the next hop to some nodes, and there is a GH node in the satellite network.

Step 2. (Line 2–3) For each piece of data $< DEST, NEXT >$ in $RT^i_{DT}$, the algorithm judges whether satellite $i$ and $NEXT$ are in the same orbit. If the condition is satisfied, jump to Step 3; if the condition is not satisfied, jump to Step 4.

Step 3. (Line 4–6) If the two satellites are in the same orbit, the algorithm further judges whether satellite $i$ and $NEXT$ are adjacent. If the condition is satisfied, this means that satellite $i$ can send packets to the next hop, and this piece of data will not cause a GH node, and the system can continue to the next piece of data. If the condition is not satisfied, this means that the next hop cannot be reached and there is a GH node in the satellite network.

Step 4. (Line 7, 11) If the two satellites are not in the same orbit, the algorithm further judges whether satellites $i$ and $NEXT$ are in adjacent orbits and adjacent to each other. If the condition is satisfied, jump to Step 5 to further judge whether it is in the link disconnected area. If the condition is not satisfied, this means that the next hop cannot be reached and a GH node exists in the network.

Step 5. (Line 8–10) The algorithm judges whether $LAT^i$ or $LAT^j$ is bigger than the polar region boundary latitude. If the condition is satisfied, this means that the next hop cannot be reached and a GH node exists in the network. If the condition is not satisfied, this means that satellite $i$ can send packets to the next hop, and this piece of data will not cause a GH node.

---

**Algorithm 2** GH detection algorithm

---

Input: $Pos_{DT}$, $LSA_{DT}$ *and* $RT_{DT}$
Output: if_Grayhole_exist
Start

1.    if $\forall(ID^i_{orb} \parallel ID^i_{sat})$ in $RT.DEST$:
2.        For each piece of data $< DEST, NEXT >$ in $RT_{DT}$
3.        if $ID^i_{orb} = NEXT.ID_{orb}$:
4.            if abs($ID^i_{sat} - NEXT.ID_{sat}$) = 1 or abs($ID^i_{sat} - NEXT.ID_{sat}$)= $M - 1$:
5.                Not GH item, continue
6.            else: GH exists, break
7.        elif abs($ID^i_{orb} - NEXT.ID_{orb}$) = 1 and $ID^i_{sat} = NEXT.ID_{sat}$:
8.            if $LAT^i$ *or* $LAT^j$ $\geq$ Polar region boundary latitude:
9.            GH exists, break
10.          else: Not GH item, continue
11.        else: GH exists, break
12.        Detect the next piece of data
13.    else: GH exists, break
14.    return if_Grayhole_exist

End

---

After completing the GH detection process, if a routing table that can lead to GH is found, the ground station will use the satellite network routing information base $RIB_{DT}$ to generate the correct routing table for the satellite and upload the routing table to the satellite. If there are no routing tables that can lead to a GH, no additional operation is required for the ground station and satellite network.

## 5. Simulations and Discussions

In this section, we conduct the experimental evaluation of the proposed detection and protection framework. We first describe the simulation scenario, experimental configuration, and the main behaviors of the satellites in the routing update process. Then, we evaluate the performance and overhead of the proposed framework and analyze the experimental results.

### 5.1. Simulation Setup

To evaluate the effectiveness of the detection and protection framework proposed in this paper, we use a computer to simulate the ground station (digital twin). At the same time, we build a satellite network simulation scenario based on the Iridium constellation [13], which is widely used in the literature, and regard it as the physical twin. The satellite network parameters are shown in Table 5. In this paper, the region with a latitude higher than 70 degrees is set as the polar region. In polar regions, the relative speed between satellites in different orbits is too fast to build stable inter-satellite links in adjacent orbits. Therefore, we only consider inter-orbital links (IOLs) in the same orbit in the polar region. In this paper, the STK 11.6 simulation tool is used to obtain the propagation delay of communication links and satellite position information. In addition, we assume that the sensor can provide correct data about the satellite network. We also assume that the secure communication protocol can ensure the correctness of data transmission. At the same time, we import the satellite network orbit data files obtained from an STK simulation into the ONE v1.6.0 simulator, which is an open-source software developed by Nokia (Finland) Research Center, to simulate network communication scenarios. The scenario randomly selects nodes to send 50 kB−1 MB sized packets to other nodes and the packets are generated at the rate of one per 1–3 s. We set 10% of nodes as SHs or GHs, and evaluate the SH or GH detection methods, respectively.

**Table 5.** The network parameters of the Iridium constellation.

| Parameter Description | Value |
| --- | --- |
| Number of satellites | 66 |
| Number of orbital planes | 6 |
| Number of satellites per orbit | 11 |
| Orbital altitude (km) | 778 |
| Orbital inclination (°) | 86 |
| Adjacent orbit equatorial longitude difference (°) | 31.6 |
| Adjacent satellite latitude difference (°) | 16.4 |
| The polar region boundary latitude (°) | 70 |
| Number of ISLs for each satellite | ≤4 |

The routing discovery, path planning, fault injection, and detection experiments were performed on a computer with the following configuration: Ubuntu 16.04 (64-bit) operating system, i7-8550 CPU, and 8 GB of RAM. We use LLFI to conduct fault injection experiments. The routing programs are written in the C programming language, and the fault injection result statistics and fault detection algorithms are implemented in Python 3.7.9. In this scenario, the satellite network uses the classic link-state-oriented routing protocol DHRP [31]. The main behaviors of the satellites in different routing phases are as follows:

(a) Route discovery phase

In the route discovery phase, to reduce the number of broadcasts and update the link state of the whole network as quickly as possible, DHRP sets up a plane speaker (PS) at each orbit plane to collect the LSAs of satellites in the plane. The PSs construct link-state

advertisement sets S-LSAs. The PSs of different orbits also exchange S-LSAs to obtain the overall link states of the satellite network, build a routing information base (RIB), and build a link delay adjacency matrix. The main program at this stage is the Build_RIB described in Table 2.

(b)    Route planning phase

In the route planning phase, the PS takes the link delay adjacency matrix built in the route discovery phase as the input parameter and uses Dijkstra's Shortest Path algorithm (DSP) algorithm to calculate the routing tables for the satellite nodes in this orbit. The satellite nodes can transmit data after acquiring the routing table. The main program at this stage is the DSP described in Table 2.

*5.2. Evaluation of Detection Capability*

The generation mechanism of SH and GH nodes caused by SDC errors has been given in Section 3. This paper proposes SH and GH nodes induced by SDC errors and the detection method based on DT and routing process files for the first time. There are few detection methods and data sets for SH and GH nodes caused by SDC errors. The existing SH or GH detection methods are based on node behavior or network traffic, such as RFTrust [17], SoS-RPL [18], and CEBD [6]. Whether caused by SDC or not, the behavior of SH or GH nodes during the forwarding phase is consistent (such as dropping data packages). That is, SH (or GH) nodes caused by SDC behave in the same way as SH (or GH) nodes caused in other ways. Therefore, from the perspective of detection capabilities, the proposed method and existing SH or GH detection methods are comparable. We construct a data set based on the satellite network scenario described in Section 5.1 to evaluate the performance and overhead of the proposed method. The data set contains 100 cases with SH nodes, 100 cases with GH nodes, 100 normal cases, and 100 other cases. We obtained the detection results of existing methods by setting SH or GH nodes and simulating communication between nodes in the ONE simulator and compared them with the proposed method.

Existing detection methods for SH and GH nodes are mainly based on traffic. This method usually detects the network traffic when the satellite network carries out the actual packet forwarding after the route update is completed. In contrast, the method proposed in this paper can detect the intermediate files in the process of satellite network route update and can find the SH and GH nodes faster and timelier. In addition, if the packet does not pass through the SH or GH nodes during packet forwarding, traffic-based detection methods cannot effectively detect the fault node. Unlike existing methods, the proposed method can logically detect SH and GH nodes caused by SDC errors by obtaining the position information, link information, routing information base, routing table, and other routing update process files of satellite nodes, thus improving the reliability and availability of satellite networks.

To evaluate whether there are False Negatives (FNs) or False Positives (FPs) in this method, we recorded the experiment results and calculated the accuracy, precision, and recall rate of the results. In this example, a sample with SH/GH nodes is defined as a positive sample, and a sample without SH/GH nodes is defined as a negative sample. The definition of relevant concepts is as follows:

- TP (True Positive): During GH/SH detection, the detection algorithm detects positive samples as positive samples;
- TN (True Negative): During GH/SH detection, the detection algorithm detects negative samples as negative samples;
- FN (False Negative): During GH/SH detection, the detection algorithm detects positive samples as negative samples, that is, the model fails to identify that the sample has a GH/SH node;
- FP (False Positive): During GH/SH detection, the detection algorithm detects negative samples as positive samples, that is, the model misreports that the sample has a GH/SH node;

- Accuracy: ac = TP + TN/(TP + TN + FP + FN);
- Precision: pr = TP/(TP + FP);
- Recall: re = TP/(TP + FN).

(1) Evaluation of SH detection capability

We select 300 RIBs from the data set, which include 100 cases with SH, 100 normal cases, and 100 cases with link loss or weight change, to evaluate the accuracy of the proposed SH detection algorithm. The confusion matrix of the results is shown in Table 6. We also compare the proposed method with the following SH detection methods:

- RFTrust [17] considers packet delivery ratio, average delay, and energy consumption, and uses Random Forest and subjective logic to detect SHs.
- SoS-RPL [18] defines two features (DI-RANK and DV-RANK) to detect SHs, and features can be updated by exchanging routing graph information.
- INTI [33] estimates the reputation of the node to detect SH attacks. Reputation is the belief that nodes establish by iterations, actions, or information exchange between them.
- InDReS [34] considers QoS Metrics and uses a constraint-based specification model to detect SH attacks.

**Table 6.** The confusion matrix of SH detection results.

| | | Actual Value | |
|---|---|---|---|
| | | Positives | Negatives |
| **Predicted value** | Positives | TP:100 | FP:2 |
| | Negatives | FN:0 | TN:198 |

The comparison of SH detection capabilities is shown in Figure 16. It can be seen that the proposed method performs well in terms of accuracy (99.3%), precision (98%), and recall (100%). Therefore, it can be considered that the proposed method has a good detection ability for the SH nodes caused by SDC errors. We also note that there are two false alarms in Table 6. In the SH detection algorithm, we use two parameters $(\overline{DLY_{iol}})$ and $(\overline{DLY_{aol}})$. For satellite constellations, the delay in an inter-orbital link is usually a fixed value. However, the $DLY_{aol}$ varies with the latitude of the satellite. Therefore, $(\overline{DLY_{aol}})$ can be seen as an adjustable parameter and an important parameter for maintaining FN and FP balance. To improve our chances of detecting a SH as much as possible, we increase the $\overline{DLY_{aol}}$ appropriately. However, this leads to false alarms in the detection results.

(2) Evaluation of GH detection capability

We select 300 RTs from the data set, which include 100 cases with GH, 100 normal cases, and 100 cases with the next hop changing to other connected nodes to evaluate the accuracy of the proposed GH detection algorithm. The confusion matrix of the results is shown in Table 7. We also compare the proposed method with the following GH detection methods:

- CEBD [6] is an extensible GH detection framework, which collects and analyzes data exchanged between nodes and constructs neural-network-based behavior classifiers to distinguish Blackhole behaviors from rational behaviors.
- Other classifiers, including SVM, CART, and ID3, can also be exploited in the CEBD framework as a comparative method.

The comparison of GH detection capabilities is shown in Figure 17. It can be seen that the proposed method performs well in terms of accuracy (100%), precision (100%), and recall (100%). Therefore, it can be considered that the proposed method has a good detection ability for GH nodes caused by SDC errors.

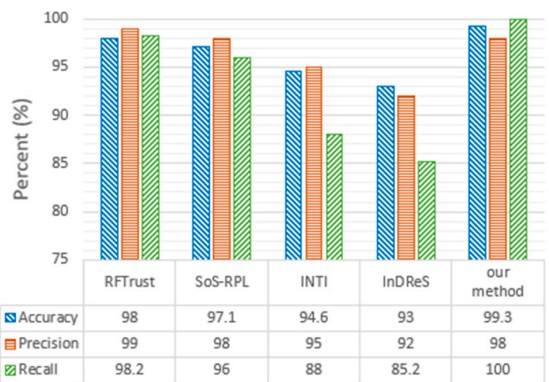

**Figure 16.** Comparison of SH detection capabilities.

**Table 7.** The confusion matrix of GH detection results.

| | | Actual Value | |
|---|---|---|---|
| | | Positives | Negatives |
| **Predicted value** | Positives | 100 | 0 |
| | Negatives | 0 | 200 |

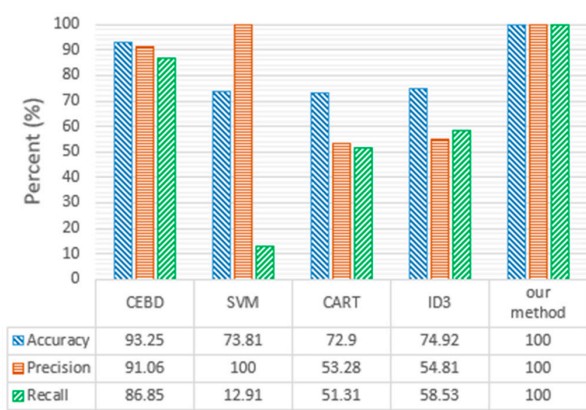

**Figure 17.** Comparison of GH detection capabilities.

*5.3. Evaluation of Performance and Cost*

(1)  Overall computing overhead

To evaluate the performance of the detection and protection framework proposed in this paper, we first discuss the computing overhead of the proposed framework in different situations. The low computing overhead operations, such as information transmission and sensor data acquisition, are not considered. Specific symbols and descriptions are as follows.

PT: physical twin (satellite network); DT: Digital Twin (ground station); NS: The normal situation; SN: The situation in which only the SH nodes caused by SDC error exist; GN: The situation in which only GH nodes caused by SDC errors exist; SGN: The situation in which both SH and GH nodes caused by SDC errors exist; GR: The cost of generating RIB; GRT: The overhead of generating routing tables for satellites; CR: The cost of checking RIB; CRT: The overhead of checking routing tables; $p$: The number of RIBs with false links that can cause SHs; $q$: The number of routing tables that can cause GHs; $r$: The number of routing tables that can cause GH nodes and do not belong to the routing tables generated by wrong RIB affected by SDC errors.

Table 8 shows the computing overheads of the satellite network and the ground station in different situations. For example, in the case of SN, the total computing overhead of DT is $K \times CR + N \times M \times CRT + p \times (GR + GRT)$. Specifically, this represents the total cost of

executing $K$ instances of RIB check, $N \times M$ instances of routing table detection, $p$ instances of RIB generation, and $p \times \frac{N \times M}{K}$ instances of routing table generation.

(2)    Total time cost of the routing update process

**Table 8.** The computing overhead of the proposed framework in different situations.

| NS | | SN (Extra Overhead) | GN (Extra Overhead) | SGN (Extra Overhead) |
|---|---|---|---|---|
| **PT** | $K \times GR + N \times M \times GRT$ | 0 | 0 | 0 |
| **DT** | $K \times CR + N \times M \times CRT$ | $P \times (GR + GRT)$ | $q \times GRT$ | $p \times (GR + GRT) + r \times GRT$ |

The total time cost consists mainly of propagation delay, transmission delay, and program running time. Compared with propagation delay, transmission delay, the running time of the program, is much shorter. Therefore, in this example, only propagation delay and transmission delay are considered to evaluate the communication time cost of the proposed framework.

The propagation delay can be expressed as $Dly_{Prop}$, as shown in Equation (7), where $Dis$ is the distance between the two sides of the communication and $Vel_{wave}$ is the propagation speed of the wave in the vacuum (About $3 \times 10^5$ km/s).

$$Dly_{Prop} = Dis/Vel_{wave} \qquad (7)$$

We use the STK tool to simulate the Iridium constellation described in Section 5.1 and obtain the link propagation delay of IOLs and AOLs, as shown in Figure 18. In this scenario, the link propagation delay of IOLs is about 13.5 ms. Considering the polar region boundary latitude, the link delay range of AOLs is about 4–12 ms. At the same time, we set up a satellite-ground communication link and obtain the link distance range between the satellite and the ground station through simulation, as shown in Figure 19. Without considering the weather and other complex factors, the link delay range between the satellite and ground station is about 3–10 ms.

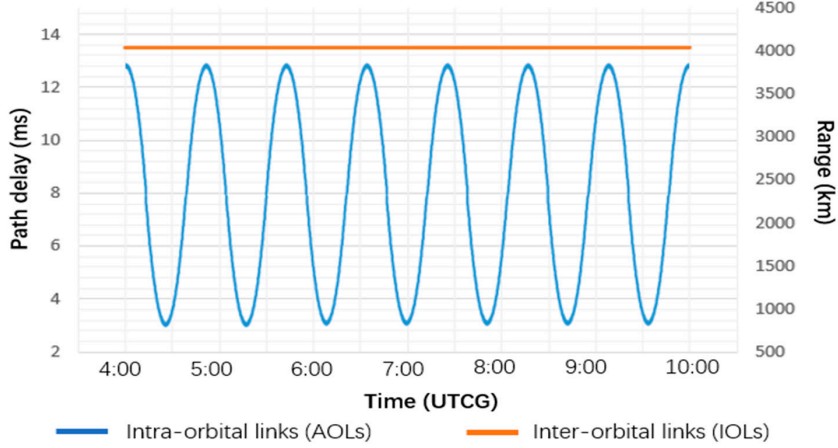

**Figure 18.** The link propagation delay of IOLs and AOLs.

The transmission delay can be expressed as $Dly_{trans}$, as shown in Equation (8), where $Size_{data}$ represents the size of the transmission data and $Tr$ represents the transmission rate of the channel.

$$Dly_{trans} = Size_{data}/Tr \qquad (8)$$

According to different transmission frequencies and communication bandwidth, $Tr$ varies from tens of K to tens of Mbps. Generally speaking, the frequency and bandwidth of the inter-satellite link, data uplink, and data downlink are different. For example, the

inter-satellite link of the MILSTAR satellite system uses 60 GHz mmWave communication with a 2.16 GHz bandwidth, and the uplink frequency and downlink frequency are about 44 GHz and 20 GHz, respectively [35,36]. For the convenience of evaluation, the transmission rates of inter-satellite link, data uplink, and data downlink in this example are set to 3 Mbps, 2 Mbps, and 1 Mbps, respectively.

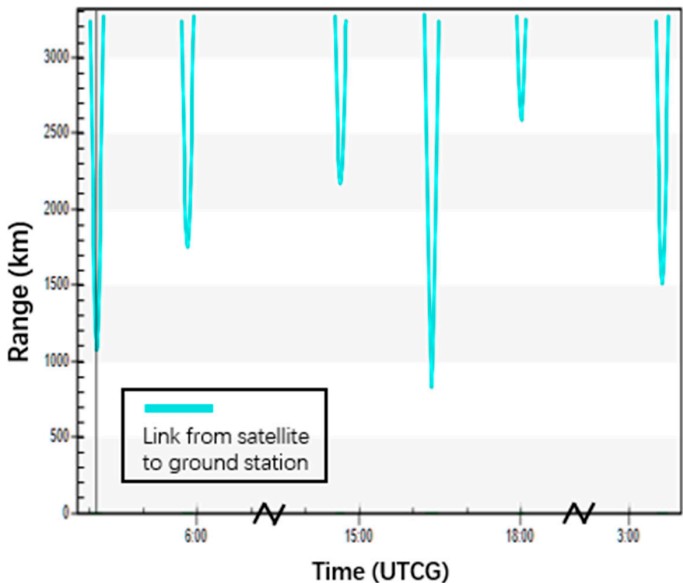

**Figure 19.** The distance range of a satellite–ground communication link.

At the same time, in the framework proposed in this paper, the five types of process files involved in the DHRP routing protocol are transmitted and stored in the form of *txt*. The size of each file, the required number of files in the satellite network, inter-satellite link transmission delay, uplink transmission delay, and downlink transmission delay are shown in Table 9, and Figure 20 contains five curves that correspond to the five types of process files involved in the DHRP routing protocol. Each curve shows the change in the transmission delay of one file with the transmission rate.

**Table 9.** Transmission delay of different files.

|  | POS | LSA | S-LSA | RIB | RT |
|---|---|---|---|---|---|
| Size (B) | 17 | 26 | 368 | 1868 | 432 |
| The number of files in PT | 66 | 66 | 6 | 6 | 66 |
| The inter-satellite link transmission delay (ms) | - | 0.07 | 0.98 | - | 1.16 |
| The uplink transmission delay (ms) | - | - | - | - | 1.73 |
| The downlink transmission delay (ms) | 0.14 | 0.21 | - | 14.94 | 3.46 |

Note: "-" means that the proposed framework doesn't need to consider this delay.

To evaluate the additional communication time cost of the detection and protection framework proposed in this paper, we take the DHRP routing protocol as an example to obtain the total routing convergence time under the NS, SN, GN, and SGN situations, respectively, as shown in Figure 21.

Under the normal situation (NS), the total routing convergence time consists mainly of the following parts: the time for collecting inter-satellite link delay (*Link delay obtaining*), the maximum delay for PS to obtain the LSA of the current orbit satellite (*LSAs to PS*), the S-LSA exchange time between PS of different orbits (*S-LSA exchange*), and the maximum delay for PS to distribute route tables for satellites in the orbit (*RT distribution*). Due to the uncertainty in PS selection in the satellite network, we consider the optimal and worst

case of S-LSA switching. The best situation is when the PS of each orbit is in the same logical position in the orbit, that is, only AOLs are used. The worst situation is that the difference between the ID of PS in the orbit 1 ($ID_{sat}^{PS1}$) and the ID of PS in the orbit 6 ($ID_{sat}^{PS6}$) is the half number of satellites per orbit ($ID_{sat}^{PS1} - ID_{sat}^{PS6} = 5$, in this case), that is, AOLs and IOLs with the maximum number of hops are used. In this case, the *Link delay obtaining* time can be calculated as $2 \times 13.5 = 27$ ms (delay of round-trip links between adjacent satellites), the *LSAs to PS* time can be calculated as $5 \times 13.5 = 67.5$ ms (link delay between PS and the satellite farthest from it), the shortest *S-LSA exchange* time can be calculated as $5 \times (9 + 0.98) \approx 50$ ms (best case scenario, time cost of 5 AOLs link transmissions), the longest *S-LSA exchange* time can be calculated as $5 \times (9 + 0.98) + 5 \times (13.5 + 0.98) \approx 123$ ms (worst case scenario, time cost of 5 AOLs and 5 IOLs transmissions), the *RT distribution* time can be calculated as $5 \times 13.5 + 1.16 \approx 73$ ms (time cost for PS to send the routing table to the satellite farthest from it), and the average *S-LSA exchange* time can be calculated as $(50 + 123)/2 = 86.5$ ms.

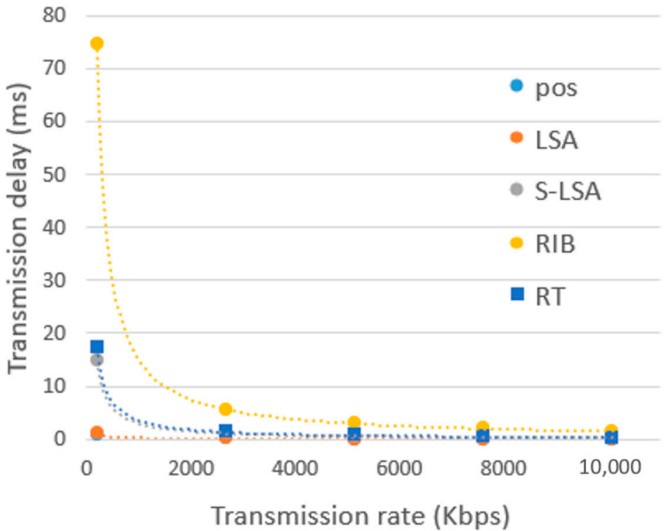

**Figure 20.** Changes in file transmission delay with transmission rate.

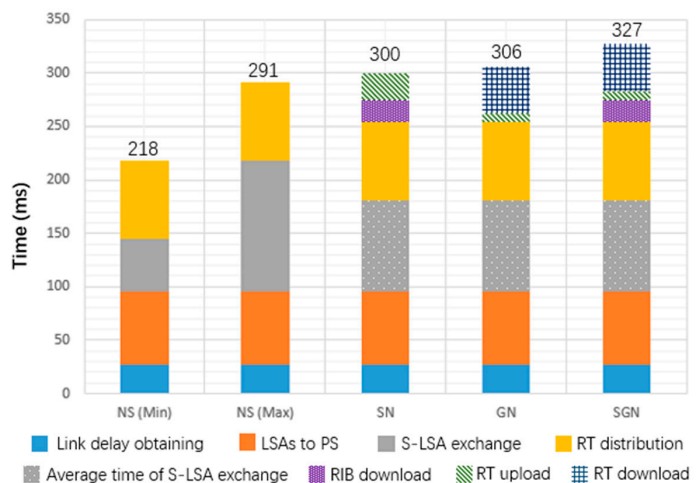

**Figure 21.** The total routing convergence time under different situations.

Under the SN situation, the additional routing convergence time consisted mainly of the following parts: the RIB download time (*RIB download*) and the routing table upload time (*RT upload*). In this example, we assume that we have detected a wrong RIB that can cause a SH. The *RIB download* time can be calculated as $14.94 + 6 \approx 21$ ms (downlink transmission delay of RIB and average propagation delay of the satellite–ground link), and

the *RT upload* time can be calculated as $11 \times 1.73 + 6 \approx 25$ ms (uplink transmission delay of RT and average propagation delay of the satellite–ground link). In other words, if an RIB of a PS has an error, the RTs of 11 satellites in this orbit need to be uploaded additionally.

Under the GN situation, the additional routing convergence time consists mainly of the following parts: the routing table download time (*RT download*) and the routing table upload (*RT upload*) time. In this example, we assume that we have detected a wrong RT that can cause a GH. The *RT download* time can be calculated as $11 \times 3.46 + 6 \approx 44$ ms (downlink transmission delay of RT and average propagation delay of the satellite–ground link), and the *RT upload* time can be calculated as $1.73 + 6 \approx 8$ ms (uplink transmission delay of RT and average propagation delay of the satellite–ground link). In other words, the PS sends the routing tables of all satellites in this orbit to the ground station, and the ground station uploads the correct RT according to the ID of the affected RT.

Under the SGN situation, if the wrong RTs are included in RTs generated according to the wrong RIB, the additional routing convergence time is the same as that of SN. If the wrong RTs are not included in the RTs generated according to the wrong RIB (this is the case in Figure 21), the additional routing convergence time consists mainly of the following parts: the RIB download time (*RIB download*), routing table download time (*RT download*), and routing table upload time (*RT upload*). In this example, we assume that we have detected a wrong RIB that can cause a SH and a wrong RT that can cause a GH. The *RIB download* time can be calculated as $14.94 + 6 \approx 21$ ms (downlink transmission delay of RIB and average propagation delay of the satellite–ground link), the *RT download* time can be calculated as $11 \times 3.46 + 6 \approx 44$ ms (downlink transmission delay of RT and average propagation delay of the satellite–ground link), and the *RT upload* time can be calculated as $1.73 + 6 \approx 8$ ms (uplink transmission delay of RT and average propagation delay of the satellite–ground link). In this case, after sending the RIB to the ground station, the PS continues to send the routing tables of all satellites in the same orbit, and the ground station uploads the correct RTs according to the ID of the affected satellites.

Figure 21 shows the routing convergence time in each case. The average time under NS (NS Avg) is 254.5 ms. Compared with NS Avg and NS Max, the time cost under SN increased by 17.6% and 3.1%; the time cost under GN increased by 20% and 5.1%; and the time cost under SGN increased by 28.2% and 12.3%, respectively. In theory, the probability of high-energy protons or galactic cosmic rays causing an SEU is 16% [1,37], and the experimental results of this article show that the probabilities of SH and GH occurrence caused by SEU are around 1% and 2%, respectively. Therefore, the probabilities of SH and GH occurrence caused by high-energy protons or galactic cosmic rays are around 0.16% and 0.32%, respectively. Therefore, the satellite network is under the NS situation most of the time. At the same time, when there are SHs and GHs caused by SEU, the additional time cost range is between 3.1% and 28.2%, which is acceptable for the satellite system as a whole.

## 6. Conclusions

SDC errors caused by SEU can affect the execution of routing programs, which may further lead to the emergence of SH and GH nodes in satellite networks, reducing the reliability and availability of the satellite networks. To solve this problem, we propose a digital-twin-based detection and protection framework for SDC-induced SH and GH nodes in satellite networks. By analyzing and detecting the digital-twin process files of the satellite network routing mechanism, the proposed framework can detect SH and GH nodes caused by SDC errors in the satellite network as early as possible before the satellite carries out actual data forwarding, allowing the failed nodes to be recovered in time. At the same time, this framework does not interfere with the normal operation of the satellite network routing mechanism when there is no SDC error or when the SDC errors do not cause SH or GH nodes to form. We evaluate the proposed detection and protection framework through experiments. Experiment results show that the detection algorithm proposed in this paper has high accuracy (99.3–100%), precision (98–100%), and recall (100%) compared to existing

approaches, and the additional time cost of routing convergence caused by this framework is relatively low (3.1–28.2%).

At present, the proposed framework mainly focuses on satellite network routing mechanisms based on link delay. We also note some shortcomings of this method, such as increasing the communication cost of satellite systems. In addition, compared to hardware-level fault injection methods or irradiation experiments, the fault coverage rate of software-based fault injection simulation methods is lower. In the future, we will improve the proposed framework by integrating other routing mechanisms, such as routing mechanisms based on link congestion and QoS. At the same time, we will consider extracting the features of the routing update process file to reduce the communication cost of the satellite system. We will also try using fault injection methods that are closer to the real radiation environment.

**Author Contributions:** Conceptualization, G.Q. and Y.Z.; methodology, G.Q. and T.Y.; software, T.Y. and G.Q.; investigation, G.Q. and Y.Q.; validation, G.Q., Y.Z. and T.Y.; result analysis, T.Y. and Y.Q.; writing—original draft preparation, G.Q. and T.Y.; writing—review and editing, Y.Z., G.Q. and Y.Q.; supervision, Y.Z.; funding acquisition, Y.Z. All authors have read and agreed to the published version of the manuscript.

**Funding:** This research was funded by the National Natural Science Foundation of China (General Program) under Grant No. 61572253.

**Data Availability Statement:** The data presented in this study are available on request from the corresponding author.

**Conflicts of Interest:** The authors declare no conflict of interest.

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
