# Peer review of "A Digital-Twin-Based Detection and Protection Framework for SDC-Induced Sinkhole and Grayhole Nodes in Satellite Networks"

_aerospace, doi:10.3390/aerospace10090788_

Round 1
Reviewer 1 Report (Previous Reviewer 1)
Dear authors,
I have thoroughly reviewed the revised version of the manuscript entitled “A Digital-Twin Based Detection and Protection Framework for SDC-Induced Sinkhole and Grayhole in Satellite Networks”, and the revised version is well-written, and the previous comments have been addressed.
The revisions related to writing style, descriptions, figures, experiments, and comparison/evaluation have been well-elaborated and have significantly enhanced the overall quality of the paper.
I would like to suggest some minor modifications that could further enhance the overall quality of the paper.
1- Section 3, end of the 1st paragraph:
I have noticed a potential concern with the statement: "At the same time, LLFI can inject faults into the routing programs by modifying source or destination register values of the targeted instruction and generates SDC, which can meet the requirements of this article."
Given the typical nature of SEU, the exact effect of fault injection cannot be determined without monitoring the program execution. Therefore, claiming that injecting faults into a targeted location will invariably generate SDC errors might be too strong of a statement. To address this, I suggest the following revision:
"At the same time, LLFI can inject faults into the routing programs by modifying source or destination register values of the targeted instruction, potentially leading to the generation of SDC. This capability aligns with the requirements of this article."
It is recommended to apply similar modifications to all similar cases throughout the paper.
2- Section 3.1., beginning of the 2nd paragraph:
I suggest clarifying the meaning of the terms SW, HW, and OS for better understanding. The current sentence states:
I propose the following revision:
"SEU may cause four results in program execution: Masked, SW-Detected (indicating errors detected by software-level), OS/HW-Detected (indicating errors detected by the operating system or hardware-level), and SDCs [22]."
3- Section 3.2., beginning of the 1st paragraph:
The authors provided information about what Control-Flow Graphs (CFG) depict, but it would be beneficial to also include a concise definition of CFG itself. You have covered the depiction aspect, but not explicitly defined CFG.
For example, a succinct definition could be: "CFG is an abstract representation of all possible execution paths of the program flow, formed by linking basic blocks using directed edges (branches)."
4- Figure 14
In Figure 14, the sequential numbering of actions in the framework is clear. However, I am confused about the numbering of actions 6/10, 7/11, and 8/12.
No further comments …
The revised version is well-written, but minor editing of English language is required.
Author Response
Please see the attachment.

Reviewer 2 Report (New Reviewer)

Author Response
Please see the attachment.

Reviewer 3 Report (New Reviewer)
This paper presents a digital-twin-based detection framework for SDC-induced Sinkhole and Grayhole in satellite networks. The proposed framework is evaluated using fault injection in the context of a simulated satellite constellation.
While the study addresses a significant problem of detecting SDC (Silent Data Corruption) in satellite networks, the problem is not new. It has been the subject of multiple studies and analyses in academia and the space industry. Also, the use of fault injection to induce faults (including SDC) and assess their implications on system availability and reliability has been explored for decades. In this context, the work presented is somewhat incremental. While the digital twins were successfully applied in multiple domains (e.g., cyber-infrastructure supporting power grid), their use for detecting SDC in a satellite network is relatively less explored. In this context, the work presents a novel approach.
Regardless, the paper raises several questions the authors should address.
1. In the Introduction, you stated: "By carrying out 10000 fault injection experiments and SDC error analysis for each program, we establish a satellite network threat model under SEU."
I am not convinced that 10000 fault injection experiments are enough to characterize SDC caused by SEU. SDCs are rare events; hence, more comprehensive fault injection experiments may be required. You should explain why you consider those 10000 faults sufficient in your study's context.
I am not clear why you use the term "threat model" (typically used in a security context) rather than "fault/error model" (typically used when analyzing reliability and availability). If you wanted to include an analysis of SDCs due to malicious attacks, you should clearly state this in the paper. If so, you should inject transient faults and representative security attacks or at least provide meaningful arguments that malicious actors could cause the consequences of injected faults.
You must be clear in your assumptions on fault models because those impact the fault injection strategy, error/failure detection, and system/application recovery. I would suggest avoiding using the term "threat model" and "malicious faults" and sticking with the "fault model."
2. Further, in the Introduction, you say: "Experiment results show that the accuracy of the proposed detection method is 98%-100%, and the additional time cost of routing convergence caused by the proposed framework is 3.1%-103 28.2%.
It would help if you clarified that those figures on accuracy are valid in the context of your experimental settings (e.g., use of a simulated environment). You should discuss the limitation of your approach and the experimental study.
3. In Section 3, you stated: "LLFI can inject faults into the routing programs by modifying source or destination register values of the targeted instruction and generates SDC, which can meet the requirements of this article."
It is somewhat unclear how you select a specific instruction to be injected.
4. In Section 4.2, you stated: "The experimental results of this article show that the probabilities of a Sinkhole or Grayhole occurrence 468 caused by SEU are around 1% and 2%, respectively." It would help if you clarified that all measurements and calculations in sections 3.2 and 3.3 are specific to your experimental settings. To generalize the results to real-world systems, you may need more evidence.
Overall the paper is easy to follow, even for non-experts. The analysis provides valid arguments to support the results/conclusions. Some more precision in describing the findings is required.
The paper is somewhat repetitive (in particular, the first half), i.e., the same things are stated repeatedly, sometimes using different words. You should remove unnecessary repetition.
Author Response
Please see the attachment.

Reviewer 4 Report (New Reviewer)
The main research motivation of this study is the development of satellite-terrestrial integrated networks and existing network resource management schemes are not satisfactory. The authors aim to solve the multi-domain network slicing problem, which is an significant research topic. Previous studies are mainly about terrestrial network slicing and satellite-terrestrial integrated network slicing is less considered in the literature, which is the research gap this study addresses. The proposed multi-sided ascending-price auction approach is innovative and novel, and it is validated effective with both theoretical analysis and numerical experiments. The manuscript is well-written with no language issues detected. I would recommend accept after some minor revisions.
1. The authors simplified the heterogeneous links, e.g., satellite-to-ground and inter-satellite links, with the bandwidth only. It would be more challenging when more metrics are involved, e.g., outage probability, bit error rate (BER), and channel capacity. More discussion and references are suggested:
* Outage probability of 3-D mobile UAV relaying for hybrid satellite-terrestrial networks
* Outage Probability and Average BER of UAV-Assisted Dual-Hop FSO Communication With Amplify-and-Forward Relaying
* Outage probability for OTFS based downlink LEO satellite communication
2. While the authors have already considered satellite and terrestrial networks, aerial networks, e.g., low altitude platforms (LAPs) and high altitude platforms (HAPs) and maritime networks, could be an important part of future multi-domain networks. The authors should discuss whether their proposed approach can be extended to more complex scenarios.
3. One challenge of game theory-based solutions is the deployment in real-world systems. The authors should describe more specific steps of implementing their proposed solutions in practice.
Author Response
Thanks for the comments. But we are not sure that these comments are for our manuscript.
The reviewer has mentioned “The authors aim to solve the multi-domain network slicing problem, which is a significant research topic.” But this is not the topic of our manuscript. And the reviewer has mentioned “One challenge of game theory-based solutions is the deployment in real-world systems. The authors should describe more specific steps of implementing their proposed solutions in practice.” But our solution is not a game theory-based solution.
Round 2
Reviewer 1 Report (Previous Reviewer 1)
Dear authors,
I have thoroughly reviewed the revised version of the manuscript entitled “A Digital-Twin Based Detection and Protection Framework for SDC-Induced Sinkhole and Grayhole in Satellite Networks”, and the revised version is well-written, and the previous comments have been addressed.
No further comments.
Reviewer 2 Report (New Reviewer)
I greatly recommend considering the reformulation of the abstract using the passive form. sing the passive form, and avoiding "other simulations tools".
English should be improved
Author Response
Please see the attachment.

Reviewer 3 Report (New Reviewer)
The authors made acceptable modifications to the paper in addressing my comments. The paper can be accepted in its current version.
The paper should be proof read for English.
Author Response
Thank you for your helpful review!
This manuscript is a resubmission of an earlier submission. The following is a list of the peer review reports and author responses from that submission.
Round 1
Reviewer 1 Report
This paper proposed a non-intrusiveness framework to solve the problem (SDC-induced Sinkhole and Grayhole) caused by SEU in satellite networks using digital-twin. Overall, the paper is well-written and presents a clear research problem. The relevant literature has been thoroughly reviewed, and the limitations and challenges facing current methods to detect SEU-induced errors in the field have been discussed in detail. The threat model is explained, and the methodology for generating errors (SDC-induced Sinkhole and Grayhole) at the program level code using a fault injection tool (LLFI) is sound. The mathematical formulas and good figures provided in the paper enhance the understanding of the research paper. The evaluation and experimentation environment are clearly described, and the method is compared with state-of-the-art methods in the field to evaluate the accuracy of detection and the imposed performance ant cost overheads.
However, I believe there are some areas that could be improved. I have provided specific comments and suggestions in the following.
Abstract
Although the abstract appears to be well-written, it lacks clarity in communicating the effectiveness of the proposed method in identifying errors and enhancing the field, in comparison to the prior methods. To increase clarity for the reader, I suggest adding quantitative evaluation results to the end of the abstract, which will enable us to better understand the efficiency of the method. Furthermore, I recommend providing some short-info on the imposed costs and overheads. Additionally, it would be beneficial to include a description of the environmental test used in this paper, to better inform readers about the experimental methodology, simulator used, etc.
Related work
The authors present a good summary, however…
Section 2, Last paragraph...
The authors state, "The traditional methods for the detection of SDC errors can be generally divided into redundancy-based detection methods and assertion-based detection methods..”. In this section, the authors classified SDC errors detection methods into two categories, but it remains unclear which category each of the discussed methods belongs to.
In addition, it is advisable to incorporate references to new SDCs detection methods. Some of the mentioned methods are old, and there are published studies that have made improvements to some of these methods (SWIFT-R and Selective SWIFT, nZDC, gZDC, etc).
Moreover, in paper [23], the authors outline the development of a machine learning-based prediction model to anticipate the occurrence of SDC by extracting instruction features and utilizing an appropriate ML model. It raises the question of why you reference this particular ML-based technique. Does it belong under the category of replication-based or assertion-based SDC detection method?
Fault Injection (FI)
1. The authors refer to conducting a large number of fault injection (FI) experiments, as indicated by references to lines 151 and 170 “Through a large number of fault injection experiments”. However, it remains uncertain whether the 20,000 injections performed is a sufficient number to yield confident outcomes. Other literature in this field typically employs a much larger FI size than this.
The fault injection campaign here involves 10K runs per a program. However, for FI, the evaluation should include an understanding of the instruction distribution in the program and the number of fault-injection runs should be computed accordingly.
- Please consider the above and comment on the number of experiments required?
- Can you please expand on the math required to compute the 10K runs required for 99% confidence in the results?
· This paper can be helpful: “R. Leveugle, A. Calvez, P. Maistri, and P. Vanhauwaert, "Statistical fault injection: Quantified error and confidence," in 2009 Design, Automation & Test in Europe Conference & Exhibition, 2009, pp. 502-506.”
2. For both sections:
3.2. Sinkhole node generation mechanism
3.3. Grayhole node generation mechanism
The authors state" To simulate the SEU of the program, we use the LLFI tool to inject 10000 faults into the Build_RIP/DSP program in the way of the single bit upset and carry out statistical analysis on the results".
One of my main concern is regarding to the precision of the fault injection campaigns carried out by LLFI, which is essentially a high-level intermediate representation (IR) FI tool. According to various studies on software-based simulation fault injection and the accuracy of fault injection at varying levels, LLFI operates at the high-IR Level code (LLVM-IR code), which means that the gap between high-IR code and low-level assembly code cannot be ignored. As a consequence, the chances of injected faults being masked, replicated, and so forth, increase. Therefore, how do the fault injection results compare in terms of accuracy to real-world soft errors?
3. Software-based fault injection (FI) techniques, such as LLFI, face the challenge of accessing internal microprocessor components for injecting faults. What level of access does LLFI have for this purpose? How does single-bit flipping actually occur in LLFI, and where is it performed? Have you attempted to modify instruction fields, memory or register contents related to the instruction, or destination registers of the targeted instruction when injecting faults?
-For instance:
· When injecting in a load instruction, have you tried to corrupt the contents of the memory location to be read (or the contents of the destination register after the load)?
· And so on with other instruction types.
I question the coverage achieved by using LLFI. In the target device hardware there are many more flops and latches that carry bits used directly and/or indirectly for the program execution. Errors cannot be injected here and their behavior cannot be understood using this LLFI.
Fault effect classification
Section “3.1. Threat model”, line 164: “SEU may cause four results in program execution: Masked, Crash, Hang, and SDC errors”
However, as discussed in many publications in literature, the categories considered here is not complete and accurate.
- What about detected errors, undetected errors, detected and corrected errors, detected and uncorrectable errors.
Safety-critical system (such satellite system) requires richer classification taxonomy that includes additional aspects such as detected and corrected errors classification (detected error, undetected errors, etc.)
Evaluation detection capability
For evaluation of Sinkhole detection capability, the traffic-based detection methods RFTrust [31], SoS-RPL [32], INTI [33], and InDReS [34] are used.
Based on the given statement, I see that these techniques differ from the proposed one as they are traffic-based. It is unclear how the comparison between the proposed technique and the existing traffic-based detection methods was conducted. To ensure a fair comparison (avoiding biased results), the implementation of each used technique and other related factors should be carefully explained.
Regarding the evaluation of Grayhole detection capability, the same questions can be raised.
Need a demonstration of this.
Other Minor comments
1. Table 1 on page 5 (line 208) requires a title, as it currently lacks one. Please ensure that a proper title is added to the table for clarity and organization.
2. Although the figures are visually apparent, it can be challenging to follow the workflow without referring to the related text (particularly Figure 13). To enhance the reader's understanding, I propose that modifications are made by numbering the workflows in the figures. This will simplify the process for the reader in tracking and keeping track of the workflow processes.
3. Call Graph: Does this graph enhance the reader's understanding by providing additional information and clarifying the problem? I may be missing something here, but I don’t feel a call-graph helps in getting an understanding of the error generating.
4. The FIGURES 3, 5, 8, 9, and 10 are actually Control-Flow Graphs (CFG) and not basic blocks. These graphs depict the control flow execution of the corresponding function. To accurately describe these figures, it is advised to replace the existing caption with the term CFG. Additionally, it would be beneficial to include a concise explanation of what a CFG is.
5. To clarify the Digital-twin concept, the reader should reach to page 11 (line 328). For improved clarity, I recommend that the concept be explained at the beginning of the article.
6. Line 410, “… the calculation module performs the routing planning and sends the correct routing table of the affected satellite to the satellite network”. I can’t see this action in the corresponding figure!!. Please consider incorporating the mentioned action into the associated figure for improved clarity.
7. Algorithm 1&2. Could you please provide a mapping of the algorithm steps to the corresponding lines of code? While your step-by-step description is helpful, I think it would greatly enhance my understanding if I could see how each step translates into the actual implementation. For instance, in Algorithm 1 “Step 3. (Line x-y) Calculate the minimum number of hops”.
8. Section 5.1, the authors state “At the same time, we build a satellite network simulation scenario based on the Iridium constellation [28], which is widely used in the existing research,”. Can you please give an example (in literature)?
9. Line 565, “the CPU is i7-8550, and the memory is 8G” > 8GB RAM.
10. Please consider selecting and adding references to a few surveys/papers that provide definitions of terms related to SEU (e.g., fault-error-failure in "Basic concepts and taxonomy of dependable and secure computing")
Reviewer 2 Report
The paper tries to address soft errors related issues in satellite applications with a digital-twin-based detection/protection mechanism.
1) The main assumption of the paper is that the CPU memory in ECC is protected but the pipeline and the computing units are vulnerable against SEUs (line 174). However, that assumption is not correct as many of today's processor vendors provide processors with lock-stepping and redundant execution mechanisms for safety-critical and high-altitude applications. An example is ARM dual-core lockstep processors which can detect transient and permanent errors in the processor pipeline effectively. Therefore, the problem that the paper is trying to solve is not an issue anymore.
Line 174: " Because the data stored in memory is usually protected by a checking mechanism (such as ECC) [27], we do not consider the impact of SEU on the static stored data. This paper mainly considers the program running errors caused by SEU in the computing unit of the processor."
2) Lack of clarity and use of qualitative description instead of quantitative description. Just a few examples:
Line 21: what is low?
‘Experiment results show that the additional time cost of routing convergence caused by the proposed framework is low.’
Line 83: How many is a lot?
“By carrying out a lot of fault injection experiments and SDC error analysis, we build a satellite network threat model under SEU”
Line 170:
“Through a large number of fault injection experiments and result analysis of routing programs, we find that SDC errors caused by SEU in satellite network routing programs are likely to cause the emergence of Sinkhole and Grayhole nodes in the network.”
What is good detection? Should use standard coverage numbers.
Line 95: we demonstrate that the proposed detection method has a good detection effect.
Reviewer 3 Report
The article A Digital-Twin Based Detection and Protection Framework for SDC-Induced Sinkhole and Grayhole in Satellite Networks is accepted with minor corrections if the authors solve these issues:
1- Please add figure to introduction section.
2- Please add comparison table to Related work section to compare this work with the rest in the Literature.
3- Improve the future work part in the conclusion section.
4- Replace References (2, 19, 20, 25, 26, and 29) with new references after 2018.
Reviewer 4 Report
The study discusses the issue of single event upset (SEU) caused by cosmic rays and high-energy particles in satellite networks, which can lead to silent data corruption (SDC) errors in routing program outputs. The research finds that SDC errors in the routing program of satellite networks can result in Sinkhole and Grayhole nodes, causing damage to satellite networks. To address this issue, the authors propose a digital-twin based detection and protection framework for SDC-induced Sinkhole and Grayhole in satellite networks. The study involves analyzing the satellite network threat model, describing the generation mechanism of Sinkhole and Grayhole, designing the required data structure of digital twins, and proposing the detection methods. The proposed detection method is experimentally validated to have high accuracy and low additional time cost for routing convergence caused by the framework.
The article presents three contributions. Firstly, a satellite network threat model under single event upset (SEU) is built through fault injection experiments and SDC error analysis. Secondly, a digital-twin based detection and protection framework is proposed for Sinkhole and Grayhole nodes induced by SDC errors. Finally, a detection algorithm of Sinkhole and Grayhole based on digital twin routing data is presented, which is experimentally validated to have a good detection effect. Additionally, the article evaluates the additional computation and time cost of the proposed detection and protection framework.
However, there are some issues.
1. As the paper mentioned, there are only a few datasets for sinkhole and gray hole nodes caused by SDC errors during the route update process, and the study constructs a dataset. However, there should be more details regarding the generated dataset, especially sinkhole and gray hole nodes. Also, the author should open-source the dataset since building the threat model of SEU is one of the claimed contributions, and it may facilitate the research community further.
2. Can the author provide some examples or quantification of how often silent data corruption cause sinkhole or gray hole in satellite network?
3. The proposed digital-twin-based detection and protection framework is based on various judging conditions of whether a sinkhole or gray hole exists in the link state advertisement, routing information base, and routing table. Why are there two false positives in the experiment of sinkhole detection results? Does it mean these rules can find some false alarm of sinkhole or gray hole? It should be explained.
4. Why is the comparison of the sinkhole and gray hole detection capabilities conducted with totally different algorithms? Are not all of them traffic-based methods that can detect sinkholes and gray holes?
5. In page 16, line 492, There is a sentence “Then the algorism judge whether…”, should it be algorithm?